*Report*

**EMBO** *reports*

# Interleukin-2-mediated NF-κB-dependent mRNA splicing modulates interferon gamma protein production

Rachel D Van Gelder [1], Nandan S Gokhale[1], Emmanuelle Genoyer[1], Dylan S Omelia[1], Stephen K Anderson [2], Howard A Young [2] & Ram Savan [1]✉

## Abstract

**Interferon-gamma (IFNγ) is a pleiotropic cytokine produced by natural killer (NK) cells during the early infection response. IFNγ expression is tightly regulated to mount sterilizing immunity while preventing tissue pathology. Several post-transcriptional effectors dampen IFNγ expression through *IFNG* mRNA degradation. In this study, we identify mRNA splicing as a positive regulator of IFNγ production. While treatment with the combination of IL-12 and IL-2 causes synergistic induction of *IFNG* mRNA and protein, defying transcription-translation kinetics, we observe that NK cells treated with IL-12 alone transcribe *IFNG* with introns intact. When NK cells are treated with both IL-2 and IL-12, *IFNG* transcript is spliced to form mature mRNA with a concomitant increase in IFNγ protein. We find that IL-2-mediated intron splicing occurs independently of nascent transcription but relies upon NF-κB signaling. We propose that while IL-12 transcriptionally induces *IFNG* mRNA, IL-2 signaling stabilizes *IFNG* mRNA by splicing detained introns, allowing for rapid IFNγ protein production. This study uncovers a novel role for cytokine-induced splicing in regulating IFNγ through a mechanism potentially applicable to other inflammatory mediators.**

**Keywords** NF-κB; Interferon Gamma; Post-transcriptional; Splicing
**Subject Categories** Immunology; RNA Biology; Translation & Protein Quality

## Introduction

Interferon-γ (IFNγ, encoded by *IFNG*) is a potent inflammatory cytokine central to the antimicrobial response through its activation of both innate and adaptive immune cell subsets (Boehm et al, 1997; De Maeyer and De Maeyer-Guignard, 1992; Schroder et al, 2004; Tewari et al, 2007). During early stages of the immune response, natural killer (NK) cells are the first producers of IFNγ, inducing protein production in a matter of minutes (Castro et al,

2018). Several cytokines can induce IFNγ in NK cells, including IL-12, IL-18, IL-15, and IL-2 (Barbulescu et al; Chaix et al, 2008; Chan et al, 1992; Nguyen et al, 2002). Combinations of these cytokines, notably IL-12 with IL-2, promote rapid production of sizeable quantities of IFNγ protein (DeBlaker-Hohe et al, 1995; Gollob et al, 1999; Hodge et al, 2002; Lusty et al, 2017; Nguyen et al, 2002; Wiedemann et al, 2021). While IL-12 is produced primarily by dendritic cells (DCs) upon sensing microbes in early infection, IL-2 is chiefly CD4+ T cell-derived, and expressed upon engagement of the T cell receptor (Perera et al, 2012; Spolski et al, 2018). Infection prompts the migration of NK cells to draining lymph nodes and secondary lymphoid structures where DCs interact with NK cells to prime them for effector functions in vivo (Lucas et al, 2007; Wiedemann et al, 2021). Early IFNγ produced by NK cells augments major histocompatibility complex II (MHC Class II) expression on APCs, allowing for subsequent T cell receptor engagement and downstream IL-2 production (Castro et al, 2018; Schroder et al, 2004; Boehm et al, 1997). Thus, NK cells in vivo may receive signals to produce IFNγ in a time-sequential manner, lending the possibility for multi-signal systems to govern *IFNG* expression.

IFNγ expression is stringently regulated to ensure its robust but transient expression, as chronic low-level IFNγ causes autoimmune disease (Hodge et al, 2014; Salerno et al, 2018a). Understanding the basis of inflammatory cytokine transcription has clarified many mechanisms of *IFNG* control, including characterization of epigenetic regulation, as well as transcriptional enhancers and their cognate transcription factors (Medzhitov and Horng, 2009; Schoenborn and Wilson, 2007; Young, 1996). However, the acute induction of IFNγ by NK cells defies transcription-translation kinetics, indicating that cells may be poised to rapidly generate IFNγ by means beyond transcriptional control. In recent years, post-transcriptional regulation of mRNA has emerged as a key process mediating the fine-tuning of inflammation (Carpenter et al, 2014; Gokhale et al, 2021; Kovarik et al, 2017; Vlasova-St Louis and Bohjanen, 2014). While the known post-transcriptional regulators of *IFNG* account for mRNA degradation and thus downregulation of IFNγ production, few mechanisms explain the swift and robust production of IFNγ in NK cells responsible for initiating antimicrobial responses. The post-transcriptional processes that

[1]Department of Immunology, University of Washington, Seattle, WA 98109, USA. [2]Cancer Innovation Laboratory, National Cancer Institute, Frederick, MD 21702, USA.
✉E-mail: savanram@uw.edu

 

positively regulate *IFNG* mRNA for rapid production of protein remain largely unknown and thus constitute the central focus of this study.

We have uncovered a novel molecular mechanism aiding in synergistic induction of IFNγ in NK cells treated with the combination of IL-12 and IL-2. We show that *IFNG* mRNA splicing is mediated by IL-2. Signaling downstream of the IL-2 receptor increases *IFNG* mRNA stability and prompts its splicing in the absence of nascent mRNA transcription, contributing to rapid translation of protein. In addition, we show that mRNA splicing induced by IL-2 is dependent upon NF-κB signaling, describing a non-canonical role for NF-κB downstream of IL-2 receptor signaling during inflammation, beyond its widely described function as a transcription factor. This study lends key insights into previously unknown post-transcriptional control of *IFNG*, with implications for understanding the broader biology of inflammatory cytokines.

# Results and discussion

## Combination of IL-12 and IL-2 synergistically induces IFNγ

Previous studies have shown that NK cells rapidly and robustly produce IFNγ upon stimulation with the combination of the cytokines IL-12 and IL-2 (Hodge et al, 2002; Ye et al, 1995). In line with these observations, we showed that treatment of CD56 + NK cells from healthy human donors with either IL-12 or IL-2 alone induced low or undetectable levels of *IFNG* mRNA expression and protein production; however, the combination of the two cytokines boosted production to levels far beyond additive effect of either cytokine alone (Fig. 1A,B). For our experiments, we primarily used a human immortal NK cell line, NK92, which phenocopied primary human NK cells in synergistic production of *IFNG* mRNA and IFNγ protein. In NK92 cells, at only 6 h post-stimulation, IL-2 and IL-12 in conjunction induced *IFNG* mRNA expression to levels almost three times greater in transcription and five times greater in protein than the additive quantity of either cytokine alone (Figs. 1C,D and EV1B). For ease of IFNγ quantification, we developed a bioassay to measure IFNγ protein expression, which correlated appropriately with our measurements of IFNγ protein through ELISA (Figs. 1A,C and EV1B). Briefly, a *Gaussia* luciferase reporter driven by GAS (Gamma activable sequence) elements from the *IRF1* promoter was stably transduced into Huh7, a hepatoma cell line. IFNγ induced STAT1 homodimers bind and activate the *IRF1* promoter-driven reporter to induce luciferase activity (Fig. EV1A). Transfer of supernatants directly from cytokine treated NK cells onto IFNγ reporter cells accurately quantified IFNγ activity through *Gaussia* luciferase production (Fig. EV1A,B).

We first assessed whether synergistic IFNγ induction in NK cells was transcriptionally mediated. We postulated that the combination of IL-12 and IL-2 might open the *IFNG* transcriptional locus for increased transcription factor binding, quickly inducing transcription of large quantities of *IFNG* mRNA, and thus accounting for robust protein output (Gollob et al, 1999; Wang et al, 1999). To this end, we investigated nascent transcription levels of *IFNG* during IL-12

stimulation alone compared to the combination of IL-12 and IL-2. To measure nascent transcription, we used 4-thiouridine (4sU), which incorporates as a nucleoside analog into actively transcribed RNA (Forero et al, 2019; Garibaldi et al, 2017). After RNA isolation, 4sU-labeled RNA was biotinylated and precipitated with streptavidin beads to determine whether rates of nascent gene transcription vary upon differential stimulation conditions. We observed that the addition of IL-2 to IL-12 treated cells resulted in a trend toward increased nascent *IFNG* mRNA induction but did not significantly induce nascent transcription of *IFNG* mRNA as compared to IL-12 treatment alone (Figs. 1E and EV1C). Our findings suggest that while transcriptional changes may partially contribute to high levels of IFNγ expression downstream of IL-2 stimulation, factors beyond transcriptional regulation likely also drive acute synergistic production of IFNγ in NK cells.

## IL-2 induces post-transcriptional stability of *IFNG* mRNA

Given that transcriptional variability alone was not sufficient to account for IL-12 + IL-2 synergism in inducing IFNγ, we wanted to determine whether active translation was necessary for the observed induction of *IFNG;* that is, whether rapid de-novo protein synthesis of another, distinct protein could account for synergistic *IFNG* induction. Using the translation inhibitor cycloheximide (CHX), whose function was validated with puromycin incorporation (Fig. EV1H), we analyzed differences in *IFNG* mRNA transcription in the absence of active translation during IL-12 versus IL-12 + IL-2 stimulation (Fig. EV1G). We found no significant difference in *IFNG* transcription between CHX- versus mock-treated cells, suggesting that nascent translation was not required for the synergistic effect of IL-12 + IL-2 stimulation. To probe whether IL-2 might globally increase protein translation, resulting in higher IFNγ output, we assayed puromycin incorporation into total actively translated proteins upon IL-12 versus IL-12 + IL-2 treatment. We observed no notable differences in puromycin incorporation between stimulation conditions (Fig. EV1I), indicating that IL-2 did not globally enhance protein translation beyond levels induced by IL-12.

Because we postulated that our observed differences in transcription and translation of *IFNG* between IL-12 versus IL-12 + IL-2 stimulated cells did not account completely for the synergism phenotype, we hypothesized that IL-2 may instead post-transcriptionally affect *IFNG* mRNA stability and induction. We observed that in the presence of IL-2, nearly 50% of *IFNG* mRNA induced upon 3 h IL-12 pre-stimulation was preserved 3 h after Actinomycin D (ActD) treatment, which blocks nascent transcription. Conversely, almost full degradation of the *IFNG* mRNA occurred during ActD treatment in absence of IL-2 (Fig. 1F). These results are comparable to the previously reported increase in *IFNG* mRNA stability during treatment with IL-12 + IL-2 (Hodge et al, 2002). We tested whether post-transcriptional stability was more globally regulated by IL-2 through investigating degradation of another inflammation-associated protein, *RELC*, as well as the RNA binding protein *HNRNPC* and housekeeping gene *GAPDH*, but found no differences in stability upon the addition of IL-2 (Fig. EV1D–F). This finding established that IL-2 prompts post-transcriptional modulation of *IFNG* mRNA, which may be responsible for its rapid and robust induction.

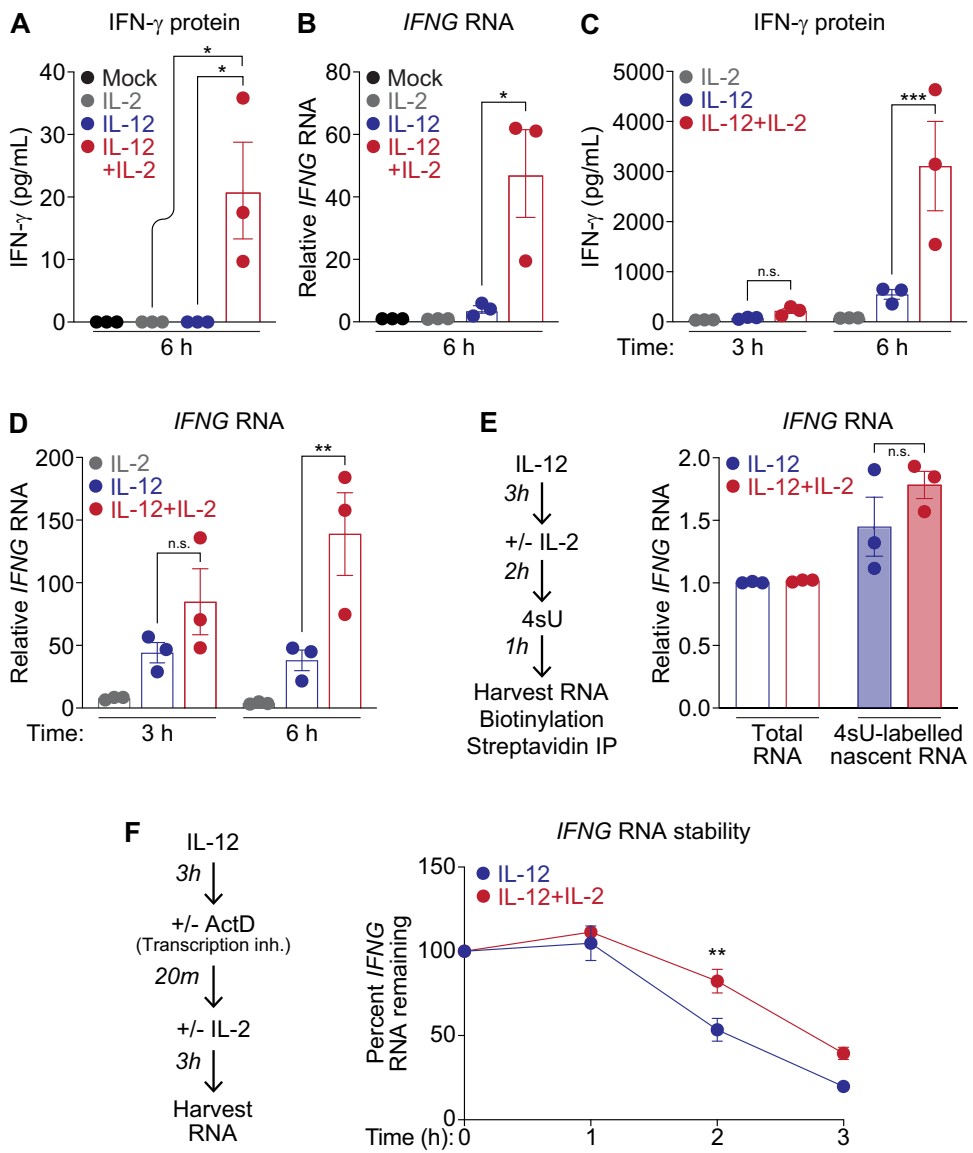

**Figure 1. IL-2 induces post-transcriptional stability of *IFNG* mRNA.**

Healthy human CD56 + NK cells were treated with IL-2 (100 U/mL), IL-12 (10 ng/mL) or both for 6 h. Supernatants and cell lysates were collected for **(A)** IFNγ protein quantification via ELISA ($p = 0.037$ for both IL-2 versus IL-12 + IL-2 and IL-12 versus IL-12 + IL-2) or **(B)** qPCR analysis of *IFNG* induction respectively (IL-12 versus IL-12 + IL-2 $p = 0.01$). *IFNG* expression normalized to *HPRT1*. NK92 cells were stimulated 3 or 6 h as in **(A)** and supernatants and cell lysates collected for **(C)** ELISA (IL-12 versus IL-12 + IL-2, 6 h stimulation, $p = 0.0009$) and **(D)** qPCR (IL-12 versus IL-12 + IL-2, 6 h stimulation, 0.0047) respectively, *IFNG* induction normalized to *HPRT1*. **(E)** qPCR analysis of total versus 4SU labeled *IFNG* for IL-12 versus IL-12 + IL-12 stimulated cells, normalized to *HPRT1* expression. **(F)** Time course of *IFNG* mRNA stability in absence of nascent transcription during IL-2 stimulation, normalized to *IFNG* levels at 3 h IL-12 treatment before addition of ActD (IL-12 versus IL-12 + IL-2, 2 h treatment, $p = 0.0097$). Data information: Data in **(A–E)** are mean ± SEM of 3 biological replicates, **(A)**, **(B)** analyzed by one-way ANOVA with Turkey's comparison test for multiple comparisons, **(C)**, **(D)** analyzed with two-way ANOVA with Turkey's test for multiple comparisons test **(E)** with ratio paired T test with Holm-Šídák method for multiple corrections; **(F)** is mean ± SEM of 3 biological replicates analyzed with two-way ANOVA with Turkey's test for multiple comparisons *$p \leq 0.05$, **$p \leq 0.01$, ***$p \leq 0.001$, n.s. is not significant. Source data are available online for this figure.

## IL-2 promotes *IFNG* mRNA splicing

We hypothesized that IL-2-dependent post-transcriptional stability of *IFNG* may be directly linked with splicing of the *IFNG* transcript. Splicing of mRNA, which often occurs co-transcriptionally, prevents transcript degradation and potentiates mRNA export and translation for subsequent protein expression (Luo and Reed, 1999; Pandya-Jones and Black, 2009). Given the more rapid degradation of *IFNG* mRNA during IL-12 stimulation alone as compared with the combination of IL-12 and IL-2, we conjectured that IL-2 might enhance *IFNG* splicing, preventing transcriptional turnover of *IFNG* mRNA and thus promoting increased translation. To test this, we first investigated intronic retention in *IFNG* mRNA using PCR primers that spanned the regions between each exon (Figs. 2A and EV2A) in both the nuclear and cytosolic fractions of NK92 cells treated with IL-12 alone or with IL-2 + IL-12. We then

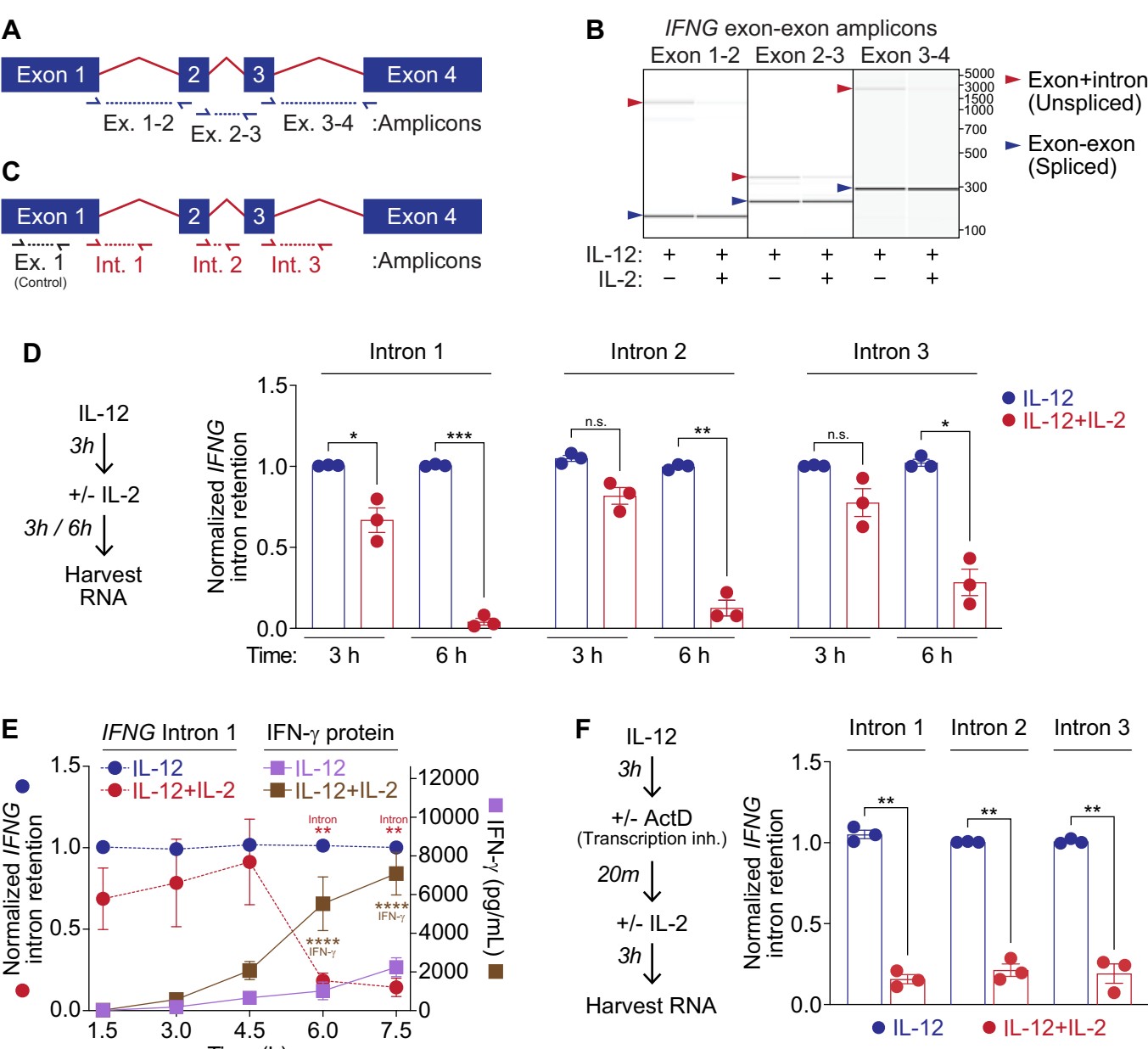

**Figure 2. IL-2 promotes *IFNG* mRNA splicing.**

(A) Primer design for PCR amplification of intra-exonic regions of *IFNG* mRNA; primers used in (B), (Fig. EV1B). (B) Bioanalyzer analysis of PCR amplification of intra-exonal regions of *IFNG* isolated from fractionated nuclei from NK92 cells stimulated with IL-12 or IL-12 + IL-2 for 6 h. (C) Schematic of SYBR probe design for qPCR amplification of intronic regions of *IFNG* in (D–F), and all further experiments quantifying intron retention (D) SYBR qPCR analysis of *IFNG* intron retention in nuclear lysates during stimulation with IL-12 or IL-12 + IL-2 for 3 or 6 h. Intron expression normalized to *IFNG* 5′UTR, representing total mature plus unspliced *IFNG* mRNA as quantified by amplification of the region spanning the 5′UTR into the coding region of Exon 1, depicted in (C) as control (Intron 1: 3 h, IL-12 versus IL-12 + IL-2, $p = 0.049$, 6 h, $p = 0.0006$; Intron 2: 6 h, $p = 0.002$; Intron 3: 6 h, $p = 0.014$). (E) SYBR qPCR analysis of *IFNG* intron retention in NK92 whole cell lysates during a time course of IL-12 or IL-12 + IL-2 stimulation for 7.5 h, normalized to the IL-12 stimulation condition and to the *IFNG* 5′UTR within each timepoint (IL-12 versus IL-12 + IL-2, 6 h treatment, $p = 0.0028$; 7.5 h $p = 0.004$). Plotted on the same graph as IFNγ protein production as measured from ELISA of supernatants also collected over the 7.5 h treatment course (IL-12 versus IL-12 + IL-2, 6 h treatment and 7.5 h treatment, $p < 0.0001$). (F) SYBR qPCR *IFNG* intron retention analysis to determine the effect of IL-2 on splicing in the absence of nascent transcription. NK92 cells were stimulated with IL-12 for 3 h to induce a pool of *IFNG* transcript, then treated with ActD to halt nascent transcription before 3 h treatment with IL-2. Data from nuclear lysates shown (IL-12 versus IL-12 + IL-2, Intron 1: $p = 0.001$; Intron 2: $p = 0.0025$; Intron 3: $p = 0.0065$). Data information: Data in (B) is representative of 3 biological replicates. Data in (D), (F) mean ± SEM of 3 biological replicates, paired T test with Holm-Šídák method for multiple corrections. Data in (E) mean ± SEM of 3 biological replicates analyzed by two-way ANOVA with Turkey's test for multiple comparisons *$p \leq 0.05$, **$p \leq 0.01$, ***$p \leq 0.001$, ****$p \leq 0.0001$, n.s. is not significant. Source data are available online for this figure.

visualized resulting amplicons by agarose electrophoresis (Fig. EV2B) or bioanalyzer (Fig. 2B). Remarkably, data from the nuclear fraction of treated cells revealed that the addition of IL-2 to IL-12-treated cells reduced intron-retaining *IFNG* mRNA transcripts, suggesting increased splicing occurred during IL-2 stimulation. We quantitatively validated this result by RT-qPCR using probes that spanned each of the three exon-intron regions and found the splicing phenotype particularly stark at the 6-hour stimulation timepoint (Fig. 2C,D). When normalized to mock stimulated cells, 30–40% of *IFNG* mRNA transcripts from IL-12 stimulated cells contained *IFNG* intron 1, while intron retaining transcripts comprised under 10% of the total *IFNG* mRNA pool during stimulation with the combination of IL-12 and IL-2 (Fig. EV2D). We sought to kinetically correlate *IFNG* mRNA splicing to increased IFNγ protein production observed during the combination of IL-12 and IL-2 stimulation to strengthen the association between mRNA processing and robust IFNγ expression. Over a time-course of 7.5 h, we found that significant *IFNG* mRNA splicing, beginning at 6 h post stimulation, coincided exactly with the induction of synergistic IFNγ protein production, when differential IFNγ expression in IL-12 versus IL-12 + IL-2 stimulated cells widened appreciably (Fig. 2E). This finding cemented the association between splicing and protein expression, suggesting that splicing correlates with increased IFNγ protein production.

To ensure that the effect of IL-2 on *IFNG* mRNA splicing occurred post-transcriptionally, we tested the ability of IL-2 to enhance splicing out of *IFNG* introns in the absence of nascent transcription using the transcriptional inhibitor ActD. During treatment with ActD, IL-2 induced splicing of *IFNG* in all three introns, confirming the phenomenon as a post-transcriptional regulatory mechanism (Fig. 2F). Using 4sU intercalation, we also found that nascently transcribed *IFNG* mRNA is spliced to a similar extent between 5- and 6-h stimulation as total transcribed *IFNG* mRNA over the course of 6 h stimulation (Fig. EV2E). This data suggests that intron-containing *IFNG* mRNA may make up a relatively larger portion of the total pool of *IFNG* mRNA than is suggested by imaging in Fig. 2B, as saturation of the spliced pool of *IFNG* mRNA plus rapid turnover of unprocessed transcript may dilute the population of intron containing *IFNG* mRNA. We additionally examined the effect of CHX-mediated translational halt on IL-2 induced *IFNG* mRNA splicing and found that RNA processing occurred independently of translation (Fig. EV2F).

Lastly, we analyzed the effects of IL-12 and IL-2 stimulation on splicing of the housekeeping gene *ACTB* to determine whether the enhanced splicing effect occurred globally (Fig. EV2G). Though there was some increased splicing in the second intron of the gene, overall IL-2 did not induce intron excision in *ACTB*, suggesting that the post-transcriptional effect of IL-2 does not apply to all cell transcripts. We conclude that a contributing role of IL-2 in inducing synergistic IFNγ is post-transcriptional splicing of *IFNG*, conferring increased mRNA stability and heightened protein production on an abbreviated timescale.

## IL-2-mediated splicing of *IFNG* mRNA is NF-κB dependent

Having characterized IL-2's effect on *IFNG* stabilization and splicing, we aimed to elucidate the mechanism downstream of IL-2 receptor (IL-2R) signaling responsible for IL-2's post-transcriptional effects. We

hypothesized that post-transcriptional effects might be mediated by ERK1/ERK 2 phosphorylation downstream of Ras/Raf activation upon ligation of IL-2R (Fig. EV3A) (Benczik and Gaffen, 2004; Lin and Leonard, 2019; Yu et al, 2000). Using the MEK1/2 phosphorylation inhibitor PD98059 we blocked ERK phosphorylation and then evaluated *IFNG* transcription, protein production and splicing. PD98059 treatment did significantly curb induction of *IFNG* mRNA and protein, in keeping with past observations (Fig. EV3B–E) (Yu et al, 2000). However, blocking the ERK signaling pathway did not affect splicing of *IFNG* (Fig. EV3F). While Ras/Raf signaling is necessary for transcriptional induction of *IFNG*, we conclude that this pathway does not control post-transcriptional processes contributing to the induction of IFNγ during IL-12 + IL-2 stimulation.

We therefore investigated whether alternative signaling downstream of the IL-2 receptor might instead control post-transcriptional regulation. PI3K signaling, thought to occur upon JAK3 phosphorylation, branches in multiple directions downstream of Akt beyond its canonical target, mTOR, which primarily controls cell proliferation and viability (Fig. EV4A) (Ali et al, 2015; Benczik and Gaffen, 2004). One of Akt's signaling targets is NF-κB, and intriguingly, we noted increased phosphorylation of p65 during the treatment of NK cells with the combination of IL-2 and IL-12 compared with IL-12 alone via immunoblot (Fig. EV4B). This contrasted with ERK1/2 phosphorylation, which was not notably altered during IL-12 versus IL-12 + IL-2 treatment (Fig. EV3B). We therefore investigated the importance of NF-κB p65 phosphorylation on IL-2 mediated *IFNG* induction. After stimulating *IFNG* transcription with IL-12, we blocked NF-κB signaling with the IκB dephosphorylation inhibitor BAY11-7802 (henceforth BAY-11) and measured *IFNG* transcription and translation (Fig. 3A,B). Stimulation with IL-12 prior to BAY-11 treatment was essential for determining the effect of NF-κB inhibition on *IFNG* induction, as inhibiting NF-κB signaling before IL-12 stimulation resulted in complete loss of *IFNG* mRNA induction (Fig. EV4C). BAY-11 treatment resulted in a reversal of IL-2 mediated *IFNG* induction and processing, inhibiting the splicing phenotype induced by IL-2 (Fig. 3A–C) and revealing a previously uncharacterized role for NF-κB in IL-2 signaling in NK cells. This phenomenon also held true during treatment with ActD, confirming that increased induction and splicing are post-transcriptional effects reliant on NF-κB signaling (Fig. 3D,E). To confirm that NF-κB dependent IL-2 mediated splicing was not a phenotype specific to the BAY-11 inhibitor, we also blocked NF-κB signaling with the NEDD8 ubiquitinoylation inhibitor MLN4924 (Milhollen et al, 2010; Soucy et al, 2009). MLN4924 inhibited induction of *IFNG* transcription and its splicing to a similar extent as BAY-11 (Fig. EV4E,F), lending validation to our studies.

Of note, we found that ActD treated cells stimulated with IL-12 plus IL-2 induce similar levels of mature *IFNG* mRNA as mock treated cells, when normalized to total degradation of mRNA within the condition (Figs. 3F and EV4D). We attribute the increased quantity of spliced *IFNG* mRNA in IL-12 + IL-2 stimulated, ActD treated cells to IL-2's post-transcriptional splicing and stabilization effect on *IFNG* mRNA. We postulate this splicing and stability prevents *IFNG* turnover and thus increases mature *IFNG* mRNA levels even during ActD treatment, a phenomenon that is reversed in the absence of NF-κB signaling. We believe this phenomenon also accounts for the slight increase in spliced RNA noted in Fig. 1F after the addition of IL-2 to ActD treated cells.

We next sought to determine whether NF-κB regulation occurs through Akt signaling downstream of PI3K activation as is

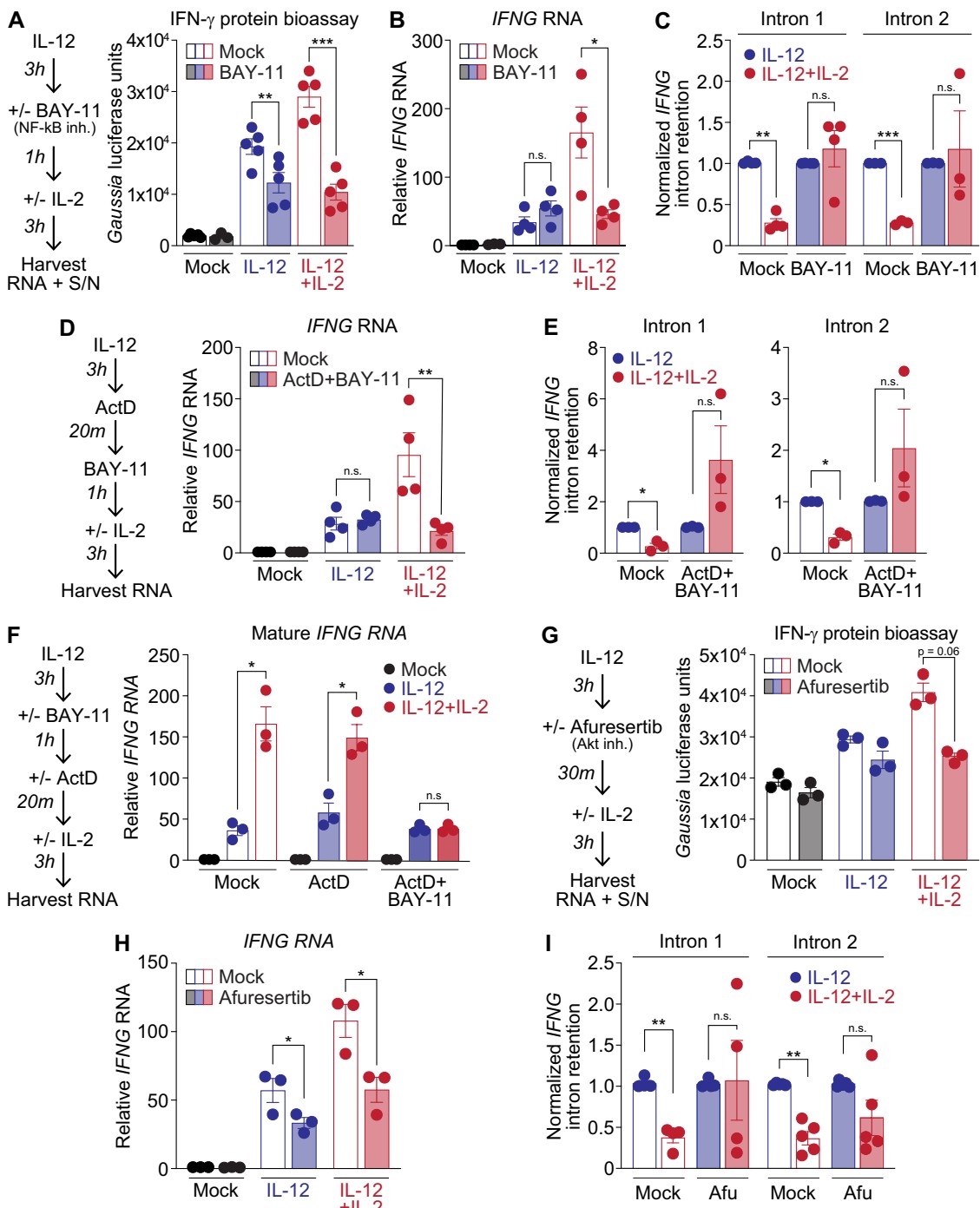

canonically described (Liu et al, 2020; Narayan et al, 2006). After inducing a pool of *IFNG* mRNA with 3 h IL-12 treatment, we blocked Akt activity with the inhibitor Afuresertib and observed that Akt inhibition restrained protein and mRNA induction to IL-12 levels even in the presence of added IL-2 (Fig. 3G–I). Interestingly, however, blocking Akt resulted in an intermediate phenotype for splicing of *IFNG* mRNA, wherein it did not appear to interrupt splicing to the same extent as BAY-11 or MLN4924 inhibition of NF-κB. We propose that while Akt signaling may partially account for the effect of NF-κB on post-transcriptional

regulation of *IFNG*, another non-canonical signaling mechanism likely also prompts phosphorylation of NF-κB p65, inducing synergistic IFNγ expression through fostering *IFNG* mRNA stability and intron excision.

## PMA promotes *IFNG* mRNA splicing in an NF-κB dependent manner

To determine whether we could replicate the effects of IL-2 in synergistic *IFNG* mRNA induction we used phorbol myristate

**Figure 3. Post-transcriptional IL-2 mediated splicing of *IFNG* is NF-κB-dependent.**

(A) Schematic for inhibition of NF-κB signaling with BAY-11 during IL-2 stimulation and *Gaussia* luciferase bioassay for IFNγ protein quantification (IL-12 comparison between treatments: $p = 0.009$; IL-12 + IL-2 comparison: $p = 0.0006$). (B) qPCR analysis of total *IFNG* mRNA induction normalized to *HPRT1* (Mock versus BAY-11 treatment during IL-12 + IL-2 treatment, $p = 0.023$) and (C) SYBR qPCR analysis of *IFNG* mRNA intron retention upon inhibiting NF-κB signaling with BAY-11 (10 μM) treatment prior to IL-2 treatment (Intron 1: Mock treatment, IL-12 versus IL-12 + IL-2, $p = 0.005$; Intron 2: Mock treatment, IL-12 versus IL-12 + IL-2, $p = 0.001$). (D) mature *IFNG* mRNA induction (Mock versus ActD+BAY-11 conditions during IL-12 + IL-2 treatment, $p = 0.007$) and (E) *IFNG* intron retention upon inhibition of both nascent transcription and NF-κB signaling prior to IL-2 treatment; normalized to *HPRT1* and to 5'UTR of *IFNG* mRNA, respectively (Intron 1: Mock treatment, IL-12 versus IL-12 + IL-2, $p = 0.048$; Intron 2: Mock treatment, $p = 0.016$). (F) *IFNG* mRNA induction normalized to *HPRT1* and to the no stimulation condition within each treatment (no treatment no stim, ActD no stim, and ActD + BAY-11 no stim); (Mock treatment, IL-12 versus IL-12 + IL-2 stimulation: $p = 0.0348$; ActD treatment, IL-12 versus IL-12 + IL-2, $p = 0.025$). (G) *Gaussia* luciferase bioassay for IFNγ protein and (H) qPCR analysis of *IFNG* mRNA normalized to *HPRT1* during inhibition of Akt activity with Afuresertib (10 μM) for 30 min prior to IL-2 stimulation (Mock versus Afu treatment during IL-12 stimulation, $p = 0.032$; during IL-12 + IL-2 stimulation, $p = 0.023$). (I) SYBR qPCR analysis of *IFNG* mRNA intron retention upon inhibiting Akt, normalized to 5'UTR of *IFNG* mRNA (Intron 1: Mock treatment, IL-12 versus IL-12 + IL-2, $p = 0.004$; Intron 2: Mock treatment $p = 0.0027$). Data information: Data in (A–I) is mean ± SEM for 3 to 5 biological replicates with (A), (B), (D), (F–H) analyzed by ratio paired T test and (C), (E), (I) by paired T test with Holm-Šídák method for multiple corrections. *$p \leq 0.05$, **$p \leq 0.01$, ***$p \leq 0.001$, n.s. is not significant. Source data are available online for this figure.

acetate (PMA), a mitogen that signals through PKC and induces NF-κB activation (Chang et al, 2005; Hellweg et al, 2006; Kemp and Bruunsgaard, 2001; Milhollen et al, 2010; Nishizuka, 1995). We hypothesized that PMA would recapitulate IL-2's effects on NF-κB signaling in synergistic induction and processing of *IFNG*. As a control, we used ionomycin, another well-characterized inducer of *IFNG* in NK and T cells, but which signals through calcium resulting in NFAT translocation to the nucleus rather than NF-κB (Macián et al, 2002; Rao et al, 1997). Upon treating cells with both IL-12 and PMA, we observed a synergistic increase in *IFNG* transcript and IFNγ protein expression that mirrored the effect of stimulation with IL-12 + IL-2 (Fig. 4A–C). The combination of ionomycin and IL-12 did not replicate this synergy (Fig. 4B,C). Given the outsized effect of the combination of IL-12 and PMA in *IFNG* mRNA induction, we probed whether PMA could promote RNA processing in the same manner that IL-2 mediated *IFNG* mRNA splicing. Treatment with PMA + IL-12 resulted in significant *IFNG* intron excision compared with IL-12 stimulation alone, to a similar extent of what we observed with IL-2 + IL-12 treatment (Fig. 4D). However, addition of ionomycin to IL-12 did not promote *IFNG* splicing, again suggesting a specific role for NF-κB in *IFNG* regulation and processing.

To validate that PMA-mediated synergistic induction and processing of *IFNG* functions primarily through NF-κB signaling, we used BAY-11 to halt NF-κB signaling after inducing a pool of *IFNG* mRNA via IL-12 treatment, but before further stimulation with PMA (Fig. 4E). We hypothesized that blocking NF-κB signaling would abrogate the synergy between IL-12 and PMA at both the transcriptional and protein levels, as well as any splicing effects. Indeed, we found that BAY-11 + PMA treated cells were unable to induce IFNγ protein or RNA to levels beyond those observed with IL-12 alone (Fig. 4E,F). In addition, BAY-11 reduced intron splicing to levels lower than those seen with IL-12 treatment alone (Fig. 4G). We therefore concluded that NF-κB signaling was required for synergistic IFNγ protein production and mRNA expression induced by the combination of IL-12 and PMA or IL-12 and IL-2.

## Intron retention is a novel post-transcriptional mechanism of *IFNG* control

In this study, we uncover IL-2 mediated intron excision as a novel post-transcriptional mechanism regulating IFNγ production.

Though it has been previously reported that the combination of the cytokines IL-12 and IL-2 stimulates rapid IFNγ production in NK cells, the mechanism by which these cytokines synergize to promote robust expression has not been elucidated (Gollob et al, 1999; Hodge et al, 2002; Wang et al, 1999; Ye et al, 1995). We found that signaling downstream of IL-2 induces *IFNG* splicing, allowing for intron-containing *IFNG* transcripts generated during IL-12 stimulation to be rapidly spliced for protein production. While we demonstrate that intron retention contributes to post-transcriptional control of *IFNG*, there are likely other mechanisms downstream of IL-2 signaling prompting the robust increase in protein observed during IL-12 + IL-2 stimulation as compared with stimulation by either cytokine alone. Of note, while our study shows that while IL-2 induces the stabilization of *IFNG* mRNA, it does not rule out that IL-2 may influence other modifications to *IFNG* mRNA. Modifications to the *IFNG* mRNA, such as $N^6$-methyladenosylation (m6A) or others, could stabilize the RNA for enhanced translatability and loading onto polysomes in a manner independent from stabilization via mRNA splicing (Roignant and Soller, 2017). Though we observed that nascent translation is not necessary for IL-2 mediated effects of *IFNG* induction and splicing, we cannot exclude this possibility as a potential mechanism resulting in synergistic induction of IFNγ protein. While we show IL-2 splicing modulates expression of IFNγ protein, we don't exclude other post-transcriptional mechanisms influencing synergistic induction.

Our study is unique in proposing a rare positive post-transcriptional regulatory control of *IFNG* mRNA. *IFNG* is known to be post-transcriptionally regulated both directly and indirectly by several microRNAs and long noncoding RNAs; however, all but one dampen rather than increase its expression (Khabar and Young, 2007; Hodge et al, 2014; Lu et al, 2011; Collier et al, 2012; Gomez et al, 2013). The RNA binding proteins (RBPs) TTP and ZFP36L2 also destabilize *IFNG* transcript through binding AU-rich elements in the 3' UTR (Freen-van Heeren et al, 2019; Salerno et al, 2018b). We have uncovered a novel pathway for induction rather than quenching of inflammatory mediator expression, which functions through splicing of retained introns within the *IFNG* mRNA transcript, adding to the repertoire of post-transcriptional modulators that contribute to increased IFNγ expression.

Intron retention has been observed in the context of several drivers of innate inflammation, including *IRF7* and *CXCL2*, in which splicing upon a second signal promotes processing of mRNA

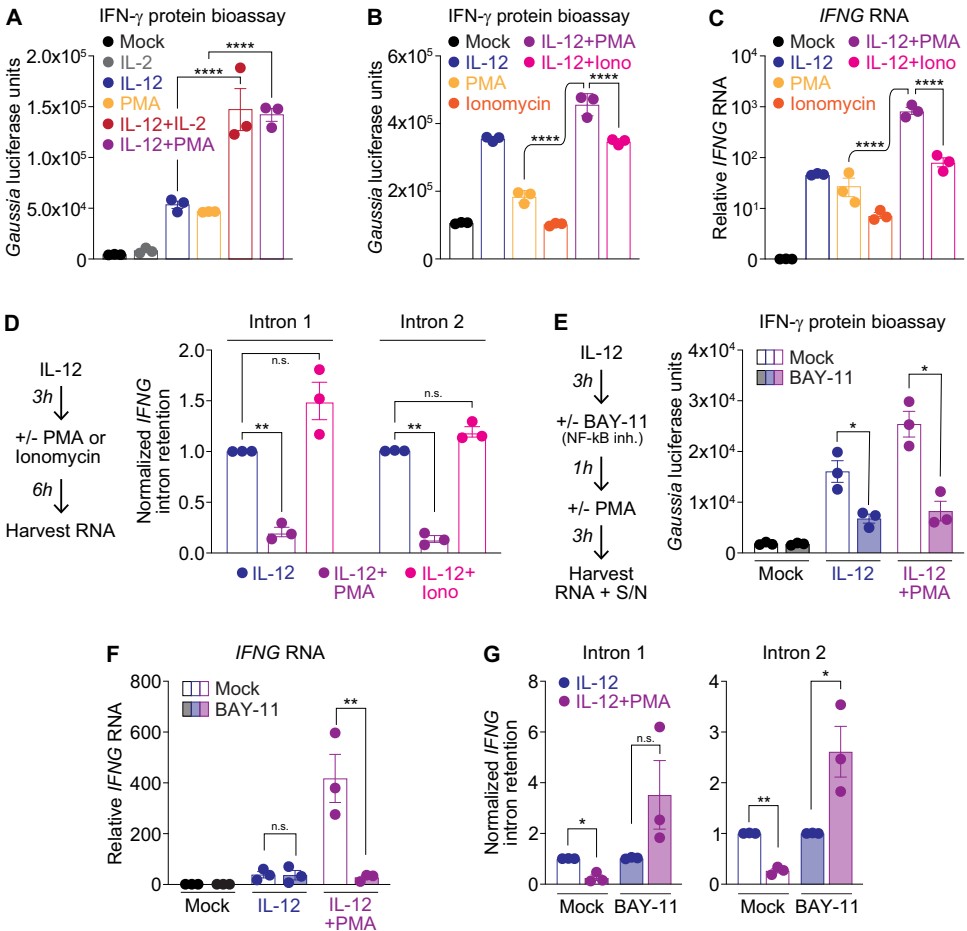

**Figure 4. NF-κB signaling is required for PMA-mediated splicing of *IFNG*.**

(A) Bioassay quantification of IFNγ protein production using Huh7 *Gaussia* luciferase IRF1 GAS(5x) reporter cells treated for 24 h with supernatants collected after 24 h NK92 stimulation with IL-2 (100 U/mL), IL-12 (10 ng/mL), PMA (10 nM), or a combination (IL-12 versus IL-12 + IL-2 and PMA versus PMA + IL-2, both $p < 0.0001$). Bioassay quantification of IFNγ protein production and (C) qPCR analysis of *IFNG* induction during stimulation with PMA (10 nM), Ionomycin (1 μg/mL), IL-12 (10 ng/mL) or a combination for 24 h (B), PMA versus PMA + IL-12, $p < 0.0001$; Iono+ IL-12 versus PMA + IL-12, $p < 0.0001$) or 6 h (C), PMA versus IL-12 + PMA, $p < 0.0001$; Iono +IL-12 *versus* PMA + IL-12, $p < 0.0001$). (D) SYBR qPCR analysis of *IFNG* mRNA intron retention of total lysates from cells treated with IL-12 (10 ng/mL), IL-12 + PMA (10 nM), or IL-12+Ionomycin (1 μg/mL) for 6 h, normalized to total mature plus unspliced *IFNG* (Intron 1, IL-12 versus IL-12 + PMA, $p = 0.0034$; Intron 2, IL-12 versus IL-12 + PMA, $p = 0.0016$). (E) *Gaussia* luciferase bioassay for IFNγ protein quantification (Mock versus BAY-11 treatment during IL-12 stimulation, $p = 0.013$; during IL-12 + IL-2 stimulation, $p = 0.025$). (F) *IFNG* mRNA transcriptional induction normalized to *HPRT1* (Mock versus ActD treatment during IL-12 + IL-2 stimulation, $p = 0.0059$) and (G) *IFNG* intron retention as amplification normalized to 5'UTR-Exon 1 amplification in the presence versus absence of 1 h NF-κB inhibitor BAY-11 (10 μM) treatment before 3 h PMA stimulation (Intron 1: Mock treatment, IL-12 versus IL-12 + IL-2 stimulation, $p = 0.04$; Intron 2: Mock treatment IL-12 versus IL-12 + IL-2 $p = 0.0039$, BAY-11+ActD treatment, $p = 0.042$). Data information: Data in (A–C) is mean ± SEM for 3 biological replicates, one-way ANOVA with Turkey's comparison test for multiple comparisons. Data in (D–G) is mean ± SEM for 3 biological replicates, (D, G) paired T test, (E, F) ratio paired T test with Holm-Šídák method for multiple corrections *$p \leq 0.05$, **$p \leq 0.01$, ***$p \leq 0.001$, ****$p \leq 0.0001$, n.s. is not significant. Source data are available online for this figure.

for its eventual translation (Bhatt et al, 2012; Frankiw et al, 2019; Green et al, 2020). In conjunction with our findings, we propose that inflammatory modulation through intron splicing regulation could be a wide-spread post-transcriptional mechanism during an immune response. We postulate that other rapidly and robustly produced cytokines such as TNFα could be governed by similar mechanisms, in which transcription of intron-containing mRNAs occurs at homeostasis or in the presence of an initial signal, but a second signal is required to induce stability and splicing for robust protein production. There is evidence to suggest that *TNFa* in mouse bone-marrow-derived macrophages is transcribed as a constitutive "pre-cursor" transcript in the unspliced form, wherein

upon MyD88/TRIF dependent LPS stimulation, splicing occurs (Hargreaves et al, 2009). In addition, in mouse T cells, preformed TNFα appears to be stabilized upon antigen stimulation even in the absence of nascent transcription, which would allow for rapid protein production (Salerno et al, 2017). Such molecular systems drastically reduce the time required to induce expression of inflammatory mediators through bypassing the need for transcriptional induction: a ready-made pool of mRNAs is available for rapid translation when specific signals stimulate processing.

It is also feasible that splicing regulation extends to other IFNγ producing cells, beyond the NK cell. CD4 + T cells are the other major producer of IFNγ protein during the inflammatory response

(Castro et al, 2018). Because IL-12 is not a primary stimulator of *IFNG* mRNA transcription in CD4 + T cells, but rather skews the lineage into an *IFNG* producing Th1 phenotype, the signals governing induction and processing of *IFNG* mRNA likely differ between NK cells and CD4 + T cells. Because T-cell receptor (TCR) signaling is typically required for CD4 + T cell production of IL-2, perhaps the role of IL-12 could be replaced by TCR signal transduction in promoting CD4 + T cell *IFNG* mRNA transcription, while delayed autocrine IL-2 signaling dependent on the initial TCR ligation might then allow *IFNG* processing for rapid translation (Spolski et al, 2018).

## NF-κB signaling downstream of the IL-2 receptor modulates post-transcriptional processes

Our findings identify NF-κB as a novel regulator of post-transcriptional processes downstream of IL-2 receptor signaling. Though NF-κB p65 is known to be activated for cytokine production in NK cells, this is thought to occur primarily downstream of activating receptors like NKG2D rather than IL-2R, which is instead canonically associated with cell survival and proliferation (Benczik and Gaffen, 2004; Kelly et al, 2002; Kwon et al, 2017). We postulate that NF-κB is likely controlling processing of *IFNG* mRNA through interactions with other RNA-binding proteins or splice factors, perhaps through acting as a scaffold to recruit proteins to intronic regions where they can facilitate mRNA processing. This seems especially pertinent given that each intron within the *IFNG* gene contains at least one NF-κB binding sequence. Indeed, p65 has been shown to serve in conjunction with DDX17 as a splice factor for HTLV (Ameur

et al, 2020). While our data suggests that NF-κB contributes to the splicing of *IFNG*, the effectors or mechanism downstream of NF-κB remain to be elucidated. Nevertheless, our observations provide new insights into how IL-2 helps induce robust IFNγ production.

Overall, this study sheds light on the mechanism by which crucial signals are stringently regulated to ensure the appropriate balance of inflammation and resolution during the immune response. The availability of pre-mRNA transcripts poised for immediate processing could serve as model for many cytokines in allowing rapid and robust production, but also quick resolution of inflammation upon termination of the signals required for post-transcriptional processing. The understanding of the post transcriptional modulation of *IFNG* mRNA can serve as a framework for investigation of the regulation of other inflammatory cytokines.

# Methods

## Cells and cell culture conditions

All cells were grown at 37 °C in 5% $CO_2$. Human NK92 cells were cultured in RPMI 1640 media supplemented with 10% fetal bovine serum (FBS), L-glutamate at 2 mM, penicillin at 100U, streptomycin at 100 μg, IL-15 at 10 ng/mL and IL-2 at 200 U/mL growth media. Before any treatment or stimulation, NK92 (ATCC: CRL-2407; CVCL_2142) cells were resuspended at a concentration of $5 \times 10^5$ cells per mL in resting media (RPMI 1640, 10% FBS, L-glutamate at 2 mM penicillin at 100U, streptomycin at 100 μg/mL with no added cytokines) overnight.

**Reagents and tools table**

| Reagent/Resource | Reference or Source | Identifier or Catalog Number |
| --- | --- | --- |
| **Experimental models** | | |
| HEK293T cells | ATCC | CRL-3216 |
| HEK293FT cells | ThermoFisher Scientific | R70007 |
| Huh7 cells | ATCC | |
| NK92 cells | Howard Young | |
| CD56 + NK cells Negatively Selected | Bloodworks | 4570-66 |
| **Recombinant DNA** | | |
| pTRIP IRF1 GAS5x hGluc-PEST | This study | |
| pMD2.G | Addgene | 12259 |
| psPAX2 | Addgene | 12260 |
| **Antibodies** | | |
| Rabbit anti-phospho ERK1/2 | Cell Signaling Technologies | 4695S |
| Rabbit anti-ERK1/2 | Cell Signaling Technologies | 4370S |
| Mouse anti-puromycin | Sigma-Aldrich | MABE343 |
| Rabbit anti-phosphorylated NF-κB p65 | Cell Signaling Technologies | 3033 |
| Rabbit anti-NF-κB p65 | Cell Signaling Technologies | 8242 |
| Rabbit anti-GAPDH | Cell Signaling Technologies | 1774S |
| Donkey anti-Rabbit IgG HRP | Jackson ImmunoResearch | 711-035-152 |
| Donkey anti-mouse IgG HRP | Jackson ImmunoResearch | 715-035-150 |

| Reagent/Resource | Reference or Source | Identifier or Catalog Number |
|---|---|---|
| **Oligonucleotides and other sequence-based reagents** | All formatted 5'–3' | |
| *IFNG* Exon 1 Forward | CAAGTTATATCTTGG CTTTTCAGCTCTGC | |
| *IFNG* Exon 2 Reverse | CTCTTTCAATTCTT CAAAATGCCTAAG | |
| *IFNG* Exon 2 Forward | AATGCAGGTCATT CAGATGTAGCG | |
| *IFNG* Exon 3 Reverse | CGAATAATTAGTC AGCTTTTCGAAGTC | |
| *IFNG* Exon 3 Forward | GAGAGTGACAGAA AAATAATGCAGAGCC | |
| *IFNG* Exon 4 Reverse | CTGGGATGCTC TTCGACCTCG | |
| *IFNG* 5'UTR Forward SYBR | GAAAGATCAG TTAAGTCCTTT | |
| *IFNG* Exon 1 Reverse SYBR | GCTTCTTTTACATATGGGTCCTGGC | |
| *IFNG* Intron 2 Reverse | GAAGGAAAGAGC ACAAACAGAGGATG | |
| *IFNG* Intron 1 Reverse SYBR | GCTACAGCAAGTCG ATATTCAGTCAT | |
| *IFNG* Exon 3 Forward SYBR | GTGGAGACCATC AAGGAAGACATG | |
| *IFNG* Exon 3 Reverse SYBR | CATAGCTTTAGCAAC TGTTAAATAGCT | |
| qRT *ACTB* Reverse | TCACCTTCACCG TTCCAGTTTT | |
| qRT *ACTB* Forward | TGACAAAACC TAACTTGCGCAG | |
| *ACTB* Exon 2 Forward | CTCACCATGGATGA TGATATCGCCG | |
| *ACTB* Intron 2 Reverse | CTGTGCAGAGA AAGCGCCCTTG | |
| *ACTB* Exon 3 Forward | CTTCTACAATGAGC TGCGTGTGGC | |
| *ACTB* Intron 3 Reverse | CAGAAGAGAGAA CCAGTGAGAAAGGGC | |
| *ACTB* Exon 4 Forward | TCCAGCTCCC TGGAGAAGA | |
| *ACTB* Intron 4 Reverse | CAGGACTTAGCTT CCACAGCACAG | |
| *ACTB* Exon 5 Forward | GCAAAGACCTGTA CGCCAACACAG | |
| *ACTB* Intron 5 Reverse | ACAGCTCCCC ACACACCACA | |
| *ACTB* Exon 6 Forward | ATCATTGCTCCT CCTGAGCGCA | |
| *ACTB* 3'UTR Reverse | GGTGTAACGCAACT AAGTCATCCG | |
| *IFNG* full 3'UTR Reverse | GGATTAAGTGAGACA GTCACAGGATATAGG | |
| *IRF1* Gene Block Forward | GAGTGTAGCCAGA TCTCCCGGGATCTCGA TATTTCCCGAAATTG | |
| *IRF1* Gene Block Reverse | CTCGAATTGGGCCC TACCCGGGATTTCGGG AAATGTAGTCTAC | |

| Reagent/Resource | Reference or Source | Identifier or Catalog Number |
|---|---|---|
| *IRF1* GAS 5x Gene Block | CCGGGATCTCGA TATTTCCCGAAA TTGATCATCGCA TTTCCCGAAATGC GAATCTGAATTT CCCGAAATCGCT TCGTAAATTTCCCG AAATCGTAGACTA CATTTCCCGAAATCCCGG | |
| Colony PCR *IRF1* Forward | CAGATCTCCCG GGATCTCGATATT TCCCGAAATTGATC | |
| Colony PCR *IRF1* Reverse | AATTGGGCCCTAC CCGGGATTTCGGG | |
| **Chemicals, Enzymes and other reagents** | | |
| 4-thiouridine | Cayman Chemical Company | 13957-31-8 |
| Afuresertib | Selleck Chemicals | S7521 |
| Advantage 2 PCR Kit | Takara | 639206 |
| Applied Biosystems Power SYBR Green PCR Master Mix | Thermo Fisher Scientific | 4367659 |
| BAY-11 | Selleck Chemicals | S2913 |
| Bio-Rad Protein Assay Kit II | Bio-Rad | 5000002 |
| Cyclohexamide | Sigma-Aldrich | 239763-M |
| Dulbecco's modified Eagle medium | Thermo Fisher Scientific | 11960069 |
| Dynabeads M-280 Streptavidin | Thermo Fisher Scientific | 11205D |
| EZ-Link HDPD-Biotin | Thermo Fisher Scientific | A35390 |
| FastDigest SmaI | Thermo Fisher Scientific | FD0664 |
| Fetal Bovine Serum | Atlas Biologicals | F-0500-A |
| Halt Protease and Phosphatase Inhibitor | Thermo Scientific | 78444 |
| HPRT1 probe | Taqman | Hs.PT.58 v.45621572 |
| Ifn gamma Human ELISA kit | Thermo Scientific | KHC4021 |
| *IFNG* probe | Taqman | Hs.PT.58.3781960 |
| InFusion | Takara | 638948 |
| Immobilon-FL PVDF Membrane | Sigma-Aldrich | IPFL00010 |
| Invitrogen Superscript IV Reverse Transcription Kit | Thermo Fisher Scientific | 18-090-050 |
| Ionomycin | Thermo Fisher Scientific | J60628 |
| Lenti-X | Clontech | 631231 |
| Midi-Prep Kit | Macherey Nagel | 740420 |
| Mini-Prep Kit | Macherey Nagel | 740588 |
| Mirus Bio TransIT-X2 Dynamic Delivery | Fisher Scientific | MIR6000 |
| MLN4924 | Cell Signaling Technologies | 85923 |
| Nucleospin RNA isolation | Macherey Nagel | 740955.25 |
| NuPAGE 10% Bis-Tris Protein Gels | Thermo Fisher Scientific | NP0301BOX |
| NuPAGE 4–12% Bis Tris Protein Gels | Thermo Fisher Scientific | NP0336BOX |
| NuPAGE MOPS SDS Running Buffer | Thermo Fisher Scientific | NP0001 |
| PD98059 | Selleck Chemicals | S1177 |
| PCR and Gel Clean Up | Macherey Nagel | 740609 |
| Pen-Strep-L-Glutamine | Corning Fisher-Scientific | MT30009CL |
| Phase Lock Gel | VWR | 10847-802 |
| Phorbol myristate acetate (PMA) | Sigma-Aldrich | P8139 |

| Reagent/Resource | Reference or Source | Identifier or Catalog Number |
|---|---|---|
| Pierce Bradford Plus Protein Assay Kits | Thermo Fisher Scientific | 23236 |
| Pierce Gaussia Luciferase Glow Assay | Thermo Fisher Scientific | 16161 |
| Precision Plus Protein Dual Color Standards | Bio-Rad | 1610374 |
| PrimeScript Reverse Transcription Kit | Takara | RR037B |
| Recombinant human IL-12 | PeproTech | 200-12 |
| Recombinant human IL-15 | Shenandoah Biotechnology | 10086100UG |
| Recombinant human IL-2 | Clinical Trial | |
| RPMI 1640 | Corning | 15-040-CV |
| SYBR Safe | Invitrogen | S33102 |
| Syringes with BD Luer-Lok TIP | BD | 302995 |
| Taqman Universal Master Mix II no UNG | Thermo Fisher Scientific | 4440048 |
| TRIzol | Thermo Fisher Scientific | 15596026 |
| UltraPure Phenol:Chloroform:Isoamyl | Sigma-Aldrich | 15593031 |
| **Software** | | |
| Adobe Illustrator | Adobe | |
| Prism 10.0 | GraphPad | |
| Image Lab | Bio-Rad | |
| BioTek Gen 5 | Agilent Technologies | |
| QuantStudio | Thermo Fisher Scientific | |
| **Other** | | |
| Synergy HT microplate reader | BioTek | |
| ChemiDoc Touch | Bio-Rad | |
| ViiA7 Real-Time qPCR system | Applied Biosystems | |
| BioAnalyzer 2100 | Agilent | |

During stimulation, rhIL-12 was used at 10 ng/mL and rhIL-2 was used at 100 U/mL. HEK293T (ATCC: CRL-1573; CVCL_0063) cells for generation of lentivirus and Huh7 (CVCL_0336) cells were cultured in Dulbecco's modified Eagle medium (DMEM) with 10% FBS, L-glutamate at 2 mM, penicillin at 100U, streptomycin at 100 μg/mL. All cell lines were tested quarterly for mycoplasma contamination and found negative. Human CD56+ primary NK cells were obtained from Bloodworks (CAT#4570-66) and rested for 1 h at a concentration of $5 \times 10^5$ cells/mL in low serum (1% FBS in RPMI) media for one hour after thawing, prior to cytokine treatments.

## Cytokine treatments and inhibitors

Cells were treated with 10 ng/mL rhIL-12, 100 U/mL rhIL-2, 10 nM PMA, and 1 μg/mL ionomycin for stimulations. The MEK1/2 phosphorylation inhibitor PD98059 (Selleck Chemicals CAT#S1177) was used at 10 μM. NF-κB inhibitors were used at the following concentrations: BAY11-7802 (Selleck Chemicals CAT#S2913) at 10 μM, MLN4924 (CST CAT#85923) at 10 μM for 1 h prior to cytokine or PMA/ionomycin treatments. The pan-Akt inhibitor Afuresertib (Selleck Chemicals CAT# S7521) was used at 10 μM for 30 min prior to stimulation. Actinomycin D (ActD) for inhibition of nascent transcription was used at 5 μg/mL for 20–30 min prior to cytokine treatments, and cycloheximide

(CHX) for inhibition of translation was used at 100 μg/mL for 15 min prior to cytokine or PMA/ionomycin treatments. No blinding was done between treated and untreated conditions.

## Gene expression analysis

RNA was isolated from NK92 cells using the Macherey Nagel NucleoSpin RNA isolation kit per manufacturer's instructions (CAT #740955). A cDNA library was then prepared using PrimeScript RT (Takara Bio CAT#RR037B) with random hexamers and oligo(dT) per manufacturer's guidelines. cDNA was amplified via qPCR using the ViiA7 qPCR system with Taqman probes and relative amplification normalized to the housekeeping gene, *HPRT1*. Primers and probes are described in the Reagents and Tools Table. Exclusion criteria was biological replicates with average technical replicate value greater than or equal to 3 standard deviations from the mean of the remaining samples.

## Quantification of IFNγ by ELISA

Supernatants from treated NK92 samples were collected at the time of RNA and/or lysate harvesting and stored at −80 °C until processing. Samples were then diluted 1:5 and processed with the IFN gamma Human ELISA kit (Thermo Scientific Cat#KHC4021) as per manufacturer instructions. Exclusion criteria was biological

replicates with average technical replicate value greater than or equal to 3 standard deviations from the mean of the remaining samples.

### *Gaussia* luciferase assay to detect IFNγ activity

All primer sequences used for cloning are provided in the Reagents and Tools Table. A gblock containing 5x repeat of the human *IRF1* promoter was amplified via PCR with the Adv2 system (Takara CAT#639206), then cloned via InFusion (Takara CAT#638948) into the lentiviral post-transcriptional regulation IP ISGF3 5x hGLUC-MODC-PEST plasmid, previously described in Schwerk et al, 2019 cut with SmaI Fast Digest enzyme. Lentivirus was generated using psPAX2 and pMD2.G with Mirus Transit X2 transfection in 293FTs. Filtered lentivirus was placed directly on Huh7 cells for 24 h. Cells were selected with puromycin, then diluted for single-cell cloning. Several single-cell clones were screened for robust range of *Gaussia* luciferase production via rhIFNγ and supernatants from NK92 cells treated with rhIL-2 and rhIL-12. For bioassay quantification of IFNγ protein via *Gaussia* luciferase readout in Huh7 IRF1 GAS(5x) cells, supernatants from stimulated NK92 cells were taken at the time RNA and lysates were collected, then frozen down at $-80\,°C$ until use. Huh7 IRF1 GAS(5x) reporters were plated at a concentration of $1.25 \times 10^5$ cells/well in a 24-well plate, then allowed to rest O/N. Media was refreshed immediately prior to adding supernatants to each well. Cells were allowed to incubate with supernatant at $37\,°C$, 5% $CO_2$ for 24 h. 100 µL of the supernatant from each well was harvested, and either frozen or prepared immediately for *Gaussia* luciferase quantification with the Pierce *Gaussia* Luciferase Glow Assay Kit (Thermo Scientific CAT#16161). Luminescence was quantified using a 96 well plate with the Synergy HT microplate reader and Biotek Gen5 analysis software.

### 4SU-labeling to determine nascent mRNA transcription

Briefly, cells under varying stimulation conditions were pulsed in the final hour of treatment with 500 µM concentration 4SU. Cells were then lysed with RA1 from the NucleoSpin RNA Kit by Macherey Nagel (CAT#740955.250) and RNA was isolated per manufacturer's instructions. Total RNA (minimum 25 µg per condition) was biotinylated with 1 mg/mL biotin-HPDP (Thermo Fisher Scientific CAT#A35390) in DMF. Preparation per 1 µg of RNA was as follows: 1 µL 10x biotinylation buffer (100 mM Tris, pH 7.5, 10 mM EDTA in $H_2O$), 7 µL RNA (1 µg diluted in RNAse free $H_2O$), 2 µL biotin-HPDP. RNA was incubated in the biotinylation solution for 2 h at RT with rotation, then phenol:-chloroform:isoamyl alcohol (Sigma-Aldrich CAT#15593031) extracted twice in phase lock tubes (VWR CAT#10847-802). RNA was precipitated with sodium acetate (12.5 µL per 100 µL aqueous phase) and pellet washed in 70% ethanol. RNA was resuspended at a concentration of 1 µg/µL. Labeled RNA was then separated using streptavidin beads. Beads were washed 3× in 0.1 M NaCl, then once in streptavidin bead binding buffer (10 mM Tris-HCL, pH 7.5, 1 mM EDTA, 2 M NaCl). RNA was denatured at $65\,°C$ for 10 min, then added to an equal volume of washed/primed streptavidin beads (Thermo Fisher Scientific CAT#12205D) and adjusted to a final volume of 200 µL in binding buffer. Streptavidin beads and RNA were rotated for 15 min at RT, then placed on a magnet for separation. Beads were washed 4× with washing buffer (100 mM Tris pH 7.5, 10 mM EDTA, 1 M NaCl, 0.1% Tween 20 in RNAse free), then eluted 2× in 10 mM DTT with 5 min incubation. RNA was then cleaned up with the NucleoSpin RNA Clean-up Kit (Macherey Nagel CAT#740948) as per manufacturer instructions. RNA was reverse transcribed and gene expression as described in the "Gene expression analysis" section.

### mRNA stability assay

NK92 cells were stimulated with IL-12 for 3 h. Subsequently, transcription was halted with Actinomycin D (5 µg/mL) for 20–30 min. Cells were treated with or without IL-2 for 4 h, and cells and supernatants were harvested at 1 h, 2.5 h, and 4 h after stimulation. RNA was isolated with the Macherey Nagel kit and analyzed via qRT PCR. Exclusion criteria was biological replicates with average technical replicate value greater than or equal to 3 standard deviations from the mean of the remaining samples.

### mRNA splicing assay

For quantification of intron retention via gel electrophoresis or Bioanalyzer, RNA was isolated from NK92 cells using the Macherey Nagel NucleoSpin RNA isolation kit as dictated by manufacture guidelines (CAT#740955.250). An *IFNG*-specific cDNA library was created from isolated RNA with the Invitrogen Super Script IV kit (ThermoFisher Scientific CAT #18-09-050) using a reverse primer specific to Exon 4 of *IFNG* (Reagents and Tools Table). cDNA was amplified using primers spanning the regions between each exon of the *IFNG* gene, provided in the Reagents and Tools Table. PCR products were run on 1% agarose gel with SYBR safe or Bioanalyzer Chip. For quantification of intron retention via qPCR, RNA was isolated from NK92 cells using the Macherey Nagel NucleoSpin RNA isolation kit as dictated by manufacture guidelines (CAT#740955.250). An *IFNG*-specific cDNA library created as described above. cDNA was then amplified via qPCR using the ViiA7 qPCR system with SYBR Green Power Mix (Thermo Scientific CAT#4367659), using primers for specific targets to regions spanning the exon-intron gap of each intron of the *IFNG* gene. Amplification of each intron was normalized to a region spanning the 5'UTR and Exon 1 of *IFNG*. Primer sequences can be found in the Reagents and Tools Table.

### Immunoblotting

Whole cell lysates were generated using RIPA buffer (150 mM NaCl, 1% NP-40, 0.5% sodium deoxycholate, 0.1% SDS, 50 mM Tris pH 7.4) supplemented with a protease phosphatase inhibitor cocktail (Thermo Scientific Cat#78444). Protein concentration was quantified with Bradford reagent (Bio-Rad Protein Assay Kit II CAT#50000002). 3–10 µg of protein was resolved by SDS-PAGE and transferred onto PVDF. Primary antibody incubation occurred overnight in with antibodies diluted in 3% BSA in TBS-T, and species specific HRP conjugated secondary antibodies for 1 h after 3× washes in TBS-T. Chemiluminescence was detected with a ChemiDoc Touch. The following antibodies were used in Western Blot analysis: anti-phospho ERK1/2 (CST CAT#4695S), anti-ERK1/2 (CST CAT#4370S), anti-puromycin (Sigma-Aldrich Cat#MABE343), anti-phosphorylated

NF-κB p65 (CST CAT#3033), anti-NF-κB p65 (CST CAT#8242), GAPDH (CST CAT#51774S).

# Data availability

No primary large data sets have been generated or deposited for this study.

The source data of this paper are collected in the following database record: biostudies:S-SCDT-10_1038-S44319-024-00324-1.

# Peer review information

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

## Acknowledgements

We thank all the members of the Ram Lab for the discussions, manuscript review, and editing. We thank Yo Okamura for technical help. This work is partly funded by grants from the National Institutes for Health: K99AI175483 (NSG); F32AI176736 (EG); R01AI175724, R21AI176442, and R21AI141823 (RS). RDVG was supported by fellowships from ITHS (5TL1TR002318-04) and UW Immunology Richard Titus fellowship. This project has been funded in part with Federal funds from the Frederick National Laboratory for Cancer Research, National Institutes of Health, under contract HHSN261200800001E. This research was supported in part by the Intramural Research Program of NIH, Frederick National Lab, Center for Cancer Research.

## Author contributions

**Rachel D Van Gelder**: Conceptualization; Data curation; Formal analysis; Investigation; Visualization; Methodology; Writing—original draft; Writing—review and editing. **Nandan S Gokhale**: Conceptualization; Data curation; Supervision; Investigation; Visualization; Methodology; Writing—review and editing. **Emmanuelle Genoyer**: Conceptualization; Investigation; Writing—review and editing. **Dylan S Omelia**: Investigation. **Stephen K Anderson**: Conceptualization; Resources; Writing—review and editing. **Howard A Young**: Conceptualization; Resources; Methodology; Writing—review and editing. **Ram Savan**: Conceptualization; Resources; Data curation; Software; Supervision; Funding acquisition; Methodology; Project administration; Writing—review and editing.

Source data underlying figure panels in this paper may have individual authorship assigned. Where available, figure panel/source data authorship is listed in the following database record: biostudies:S-SCDT-10_1038-S44319-024-00324-1.

## Disclosure and competing interests statement

The authors declare no competing interests.

# Expanded View Figures

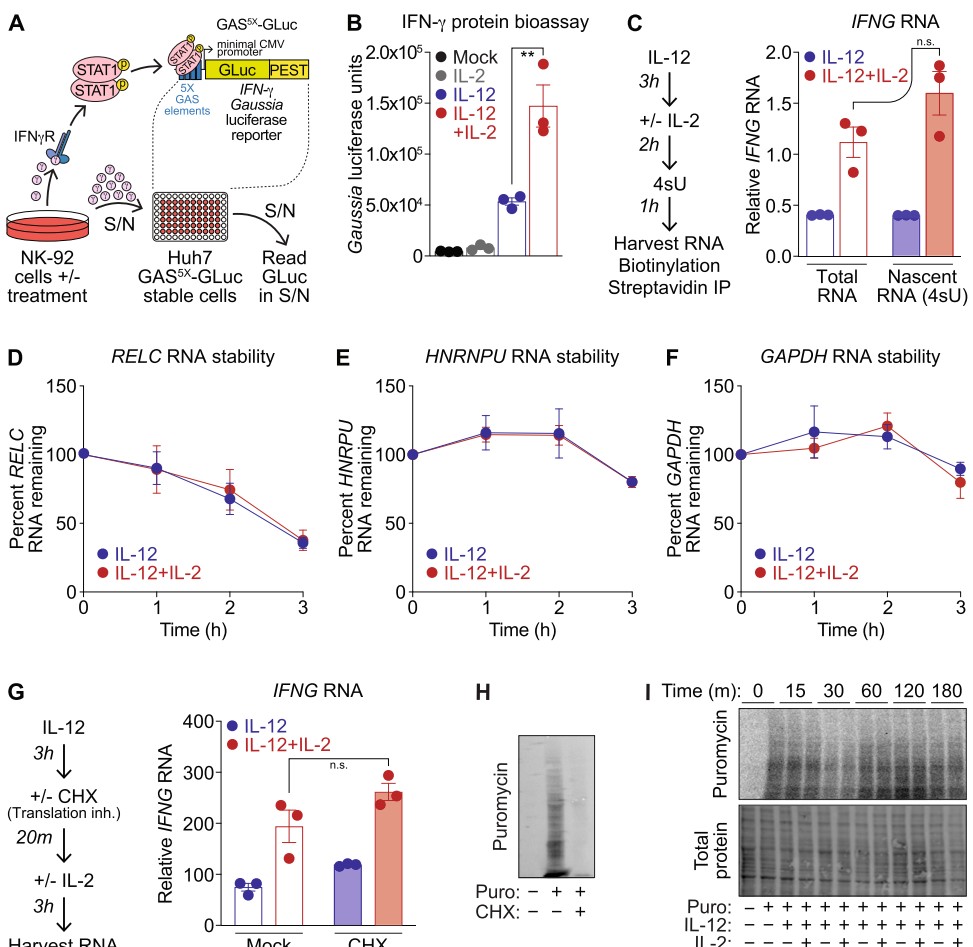

**Figure EV1. IL-2 stimulation does not globally affect transcript stability or protein translation in NK92 cells.**

(A) Schematic of IFNγ bioassay workflow for quantification of IFNγ in supernatants of treated NK92 cells via Huh7 *Gaussia* luciferase IRF1 GAS(5x) reporter cells. (B) Bioassay quantification of IFNγ protein production using supernatants from NK cells stimulated 24 h with IL-2 (100 u/mL), IL-12 (10 ng/mL) or IL-12 + IL-2 (IL-12 versus IL-12 + IL-2 stimulation, $p = 0.0037$). (C) qPCR analysis of 4SU labeled *IFNG* transcripts compared with total (4SU labeled + unlabeled) *IFNG* induction during IL-12 versus IL-12 + IL-2 treatment; normalized to *HPRT1*. Time course of (D) *RELC*, (E) *HNRNPU*, and (F) *GAPDH* mRNA stability in absence of nascent transcription during IL-2 stimulation, normalized to *RELC*, *HNRNPU*, and *GAPDH* levels at 3 h IL-12 treatment before addition of ActD, respectively. (G) qPCR analysis of *IFNG* mRNA induction during IL-2 stimulation in presence and absence of nascent translation, using cycloheximide (CHX) treatment (100 µg/mL) to halt protein synthesis. Normalized to *HPRT1* expression (H) Immunoblot depicting puromycin incorporation for cycloheximide protein synthesis halt control. (I) Immunoblot depicting puromycin incorporation in NK92 cells in NK92 cells stimulated over a time course of 0–3 h with IL-12 or IL-12 + IL-2. Total protein stain used for relative comparison of puromycin incorporation. Data information: Data in (B–G) are mean ± SEM of 3 biological replicates, (B) one-way ANOVA with Turkey's comparison test for multiple comparisons, (C) and (G) ratio paired T test with Holm-Šídák method for multiple corrections, (H) is representative of 3 biological replicates, (I) is representative of 2 biological replicates. **$p \leq 0.01$, n.s. is not significant.

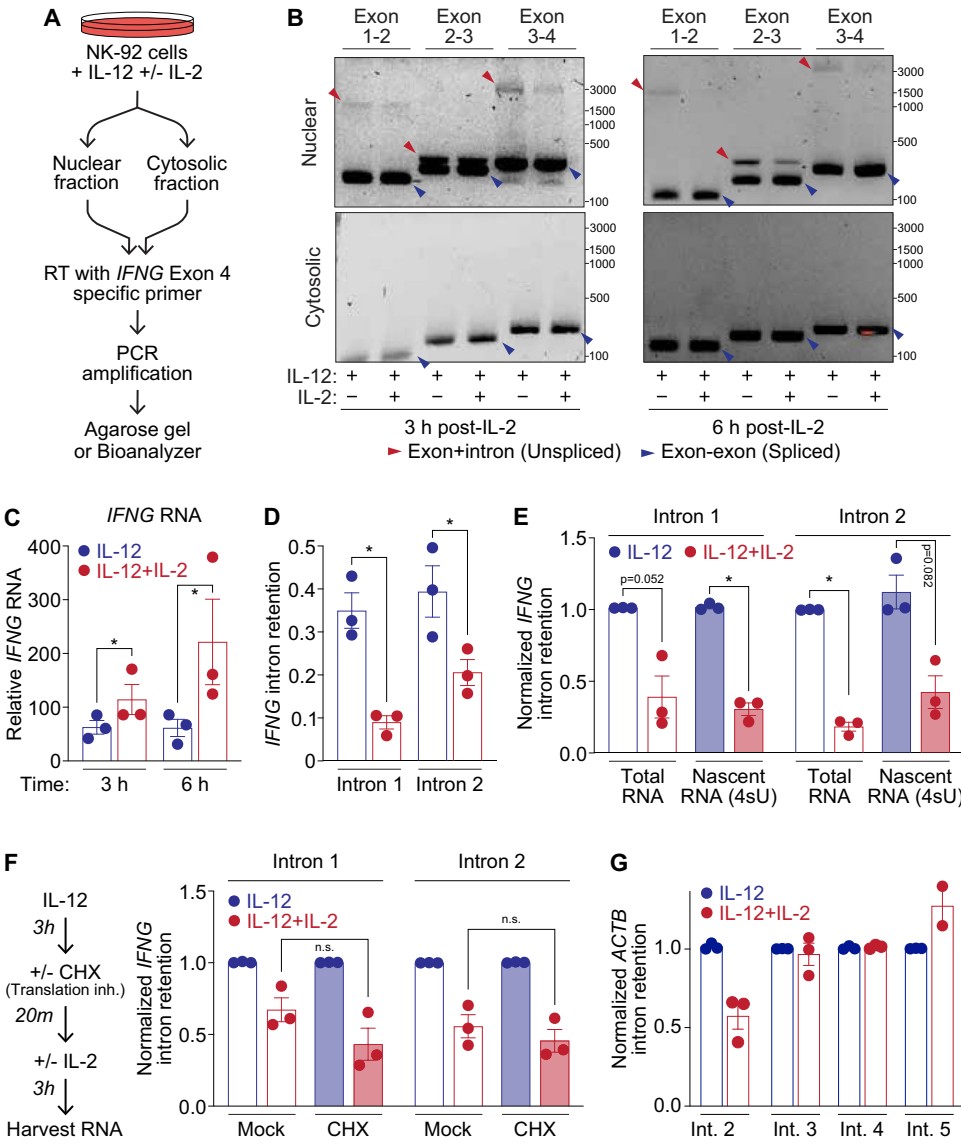

**Figure EV2.  IL-2 mediates *IFNG* intronic splicing during nascent translation and upon cycloheximide treatment.**

(A) Schematic of NK92 stimulation and processing for determining *IFNG* intron retention via PCR as seen in Fig. 2B, EV1B. (B) Image of gels depicting PCR products from amplification of intra-exonal regions of *IFNG* from nuclear and cytosol fractionated NK92 cells stimulated with IL-12 or IL-12 + IL-2 for 3 or 6 h. Blue arrows depict fully spliced products while red arrows depict amplified intronic regions. (C) total spliced *IFNG* induction (IL-12 versus IL-12 + IL-2 stimulation, 3 h treatment *p* = 0.04; 6 h, *p* = 0.044) normalized to *HPRT1* expression during stimulations outlined in Fig. 2D. (D) SYBR qPCR analysis of *IFNG* intron retention in whole cell lysates during stimulation with IL-12 for 3 h with or without subsequent 3 h IL-2 stimulation. Each stimulation condition normalized to mock treatment condition. Intron expression normalized to *IFNG* 5′UTR, representing total mature plus unspliced *IFNG* mRNA as quantified by amplification of the region spanning the 5′UTR into the coding region of Exon 1 as control (Intron 1, IL-12 versus IL-12 + IL-2 stimulation, *p* = 0.014; Intron 2, *p* = 0.025). (E) SYBR qPCR analysis of *IFNG* intron retention in nascently transcribed or total mRNA from whole cell lysates stimulated for 6 h with IL-12 or IL-12 + IL-2. Cells were pulsed with 4SU between 5- and 6-h stimulation. Intron expression normalized to *IFNG* 5′UTR (Intron 1: 4SU treatment, IL-12 versus IL-12 + IL-2, *p* = 0.013; Intron 2: Mock, IL-12 versus IL-12 + IL-2 stimulation, *p* = 0.02). (F) SYBR qPCR analysis of *IFNG* intron retention upon IL-2 treatment with and without translation inhibition via CHX (100 µg/mL) treatment. Normalized to total mature plus unspliced *IFNG* expression via 5′UTR-Exon 1 amplification. (G) Analysis of intron retention in *ACTB* during IL-12 or IL-12 + IL-2 stimulation for 6 h, normalized to total expression of *ACTB* via amplification of Exon 6/3′UTR. Data information: Data in (B) representative of 3 biological replicates. Data in (C–G) depict mean ± SEM of 3 biological replicates, where (C) is analyzed by ratio paired T test and (D–F) paired T test with Holm-Šídák method for multiple corrections *\*p* ≤ 0.05, n.s. is not significant.

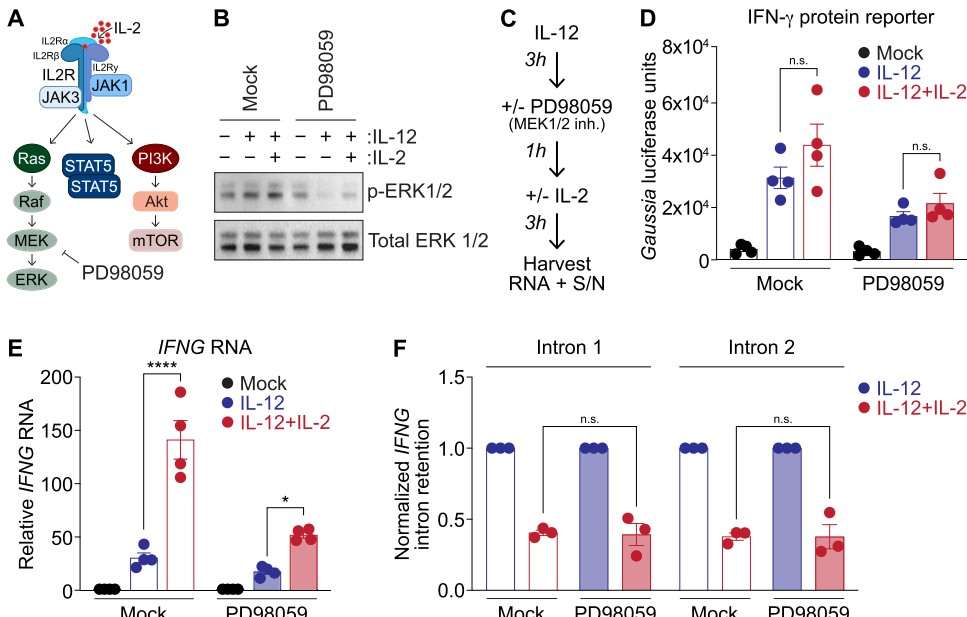

**Figure EV3.** **ERK signaling does not post-transcriptionally regulate *IFNG* mRNA processing.**

(A) Schematic of canonical IL-2R signaling. (B) Immunoblot confirming effect of inhibitor PD98059 (10 μM) on preventing phosphorylation ERK1/2 downstream of MEK1/2 inhibition during IL-12 and IL-2 treatment in NK92 cells (C) schematic of PD98059 treatment in determining effect on IFNγ induction and splicing. (D) *Gaussia* luciferase bioassay for IFNγ protein quantification (E) total *IFNG* transcript induction and (F) *IFNG* intron retention upon treating cells with PD98059 (25 μM) prior to IL-2 treatment, normalized to total spliced plus unspliced *IFNG* (Mock treatment, IL-12 versus IL-12 $p < 0.0001$; PD98059 treatment, IL-12 versus IL-12 + IL-2 $p = 0.017$). Data information: Data in (D–F) is mean ± SEM for 3 or 4 biological replicates. (B) is representative of 3 biological replicates. (D–E) analyzed two-way ANOVA with Turkey's test for multiple comparisons and (F) by paired T test with Holm–Šídák method for multiple corrections. $*p \leq 0.05$, $****p \leq 0.0001$, n.s. is not significant.

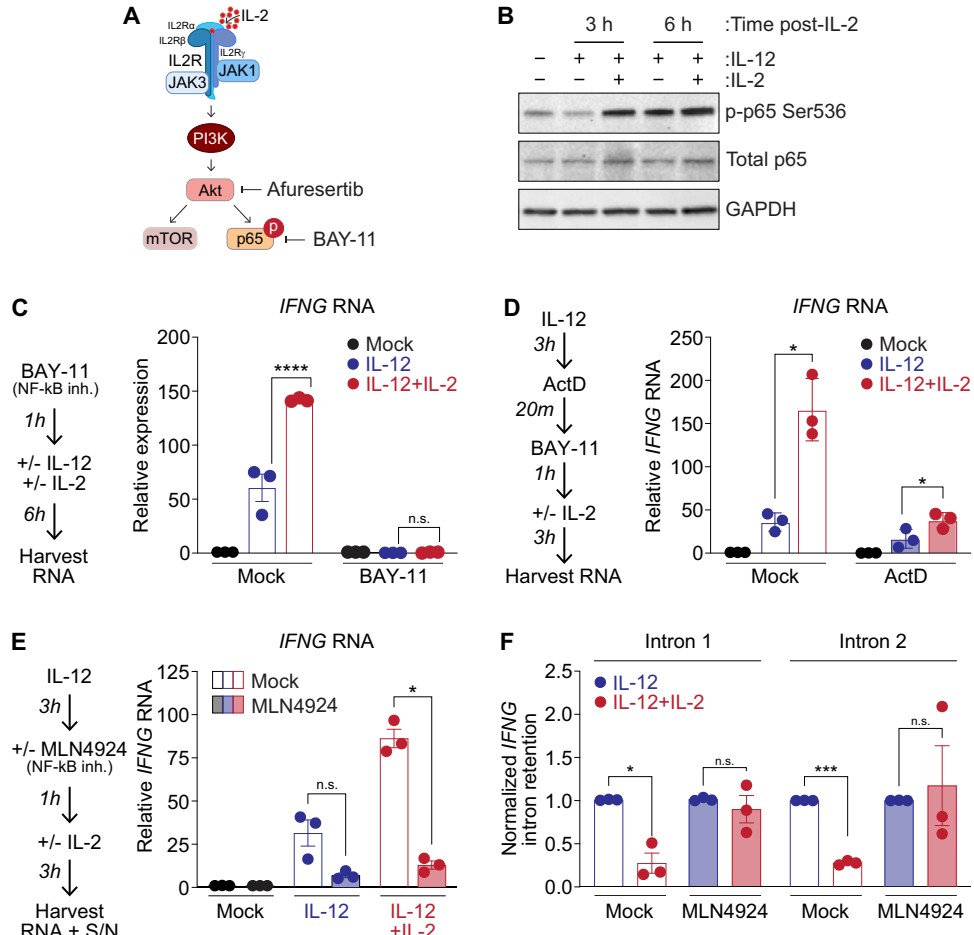

**Figure EV4. NF-κB signaling downstream of IL-2 is required for post-transcriptional regulation of *IFNG* mRNA.**

(A) Schematic depicting alternative signaling through Akt downstream of the IL-2 receptor in NK cells (B) immunoblot for total and phosphorylated NF-κB p65 NK cells treated with IL-12 or IL-12 + IL-2 for 3 or 6 h with GAPDH as loading control (C) qPCR analysis of *IFNG* induction upon pre-treatment with the NF-κB inhibitor BAY-11, prior to any stimulation (Mock treatment, IL-12 versus IL-12 + IL-2 comparison $p < 0.0001$). (D) *IFNG* mRNA induction normalized to *HPRT1* with all samples normalized to the no stimulation condition within the mock treatment to confirm functioning ActD halt of nascent transcription (Mock treatment, IL-12 versus IL-12 + IL-2, $p = 0.0397$; ActD treatment, IL-12 versus IL-12 + IL-2, $p = 0.0257$). (E) qPCR analysis of total *IFNG* mRNA induction (Mock versus MLN4924 treatment during IL-12 + IL-2 stimulation, $p = 0.015$) and (F) SYBR qPCR analysis of *IFNG* mRNA intron retention upon inhibiting NF-κB signaling with MLN4924 (10 μM), (Intron 1, Mock treatment, IL-12 versus IL-12 + IL-2 $p = 0.049$; Intron 2 Mock treatment $p = 0.0004$). Data information: (B) is representative of 5 biological replicates where 4 of 5 showed the depicted result. Data in (C–F) is mean ± SEM for 3 biological replicates with (C) analyzed by one-way ANOVA with Turkey's test for multiple comparisons, (D) analyzed by two-way ANOVA with Turkey's test for multiple comparisons, (E) analyzed by ratio paired T test and (F) by paired T test with Holm-Šídák method for multiple corrections *$p ≤ 0.05$, ***$p ≤ 0.001$, ****$p ≤ 0.0001$, n.s. is not significant.

