## [Peer Review File · EMBO Reports]

Interleukin-2-mediated NF- κ B-dependent mRNA splicing modulates interferon gamma protein production

Rachel Van Gelder, Nandan Gokhale, Emmanuelle Genoyer, Dylan Omelia, Stephen Anderson, Howard Young, and Ram Savan

Corresponding author(s): Ram Savan (savanram@uw.edu)

Review Timeline:

Submission Date:	13th May 24
Editorial Decision:	11th Jun 24
Revision Received:	23rd Sep 24
Editorial Decision:	17th Oct 24
Revision Received:	3rd Nov 24
Accepted:	8th Nov 24

Editor: Achim Breiling

Transaction Report:

Dear Dr. Savan,

Thank you for the transfer of your manuscript to EMBO reports. I have now received the reports from the three referees that were asked to evaluate your study, which can be found at the end of this email. As you will see, the referees have several comments, concerns, and suggestions, indicating that a major revision of the manuscript is necessary to allow publication of the study in EMBO reports. As the reports are below, and all the concerns need to be addressed, I will not detail them further here. Nevertheless, I think it would be important to demonstrate in the revised manuscript that altered lncRNA splicing is responsible for changes to IFN- γ protein abundance (as requested by referee #1 and #2), or to tone down related claims and discuss this in detail, including possibilities that protein level changes are regulated by mechanisms other than intron retention.

Referee #3 indicated during cross-commenting, that an elegant way to show the above would be a time-course assay demonstrating inverse correlations between intron retention and protein abundance.

Given the constructive referee comments, I would like to invite you to revise your manuscript with the understanding that the concerns of the referees must be addressed in the revised manuscript and in a detailed point-by-point response. Acceptance of your manuscript will depend on a positive outcome of a second round of review. It is EMBO reports policy to allow a single round of revision only and acceptance of the manuscript will therefore depend on the completeness of your responses included in the next, final version of the manuscript.

Moreover, please have your revised manuscript carefully edited by a native speaker before re-submission.

- 1) a .docx formatted version of the final manuscript text (including legends for main figures, EV figures and tables), but without the figures included. Figure legends should be compiled at the end of the manuscript text.
- 2) individual production quality figure files as .eps, .tif, .jpg (one file per figure), of main figures and EV figures. Please upload these as separate, individual files upon re-submission.

For papers with not more than 5 main and EV figures, we would publish your manuscript as Report. For a Scientific Report we require that results and discussion sections are combined in a single chapter called "Results & Discussion". Please do this for your manuscript, if the numbers of figures will not increase. For more details please refer to our guide to authors:

<http://www.embopress.org/page/journal/14693178/authorguide#researcharticleguide>

- 3) a .docx formatted letter INCLUDING the reviewers' reports and your detailed point-by-point responses to their comments. As

part of the EMBO Press transparent editorial process, the point-by-point response is part of the Review Process File (RPF), which will be published alongside your paper.

4) a complete author checklist, which you can download from our author guidelines (<https://www.embopress.org/page/journal/14693178/authorguide>). Please insert page numbers in the checklist to indicate where the requested information can be found in the manuscript. The completed author checklist will also be part of the RPF.

Please also follow our guidelines for the use of living organisms, and the respective reporting guidelines: <http://www.embopress.org/page/journal/14693178/authorguide#livingorganisms>

5) that primary datasets produced in this study (e.g. RNA-seq, ChIP-seq, structural and array data) are deposited in an appropriate public database. If no primary datasets have been deposited, please also state this in a dedicated section (e.g. 'No primary datasets have been generated and deposited'), see below.

The accession numbers and database should be listed in a formal "Data Availability" section (placed after Materials & Methods) that follows the model below. This is now mandatory (like the COI statement). Please note that the Data Availability Section is restricted to new primary data that are part of this study. This section is mandatory. As indicated above, if no primary datasets have been deposited, please state this in this section

Data availability

8) Regarding data quantification and statistics, please make sure that the number "n" for how many independent experiments were performed, their nature (biological versus technical replicates), the bars and error bars (e.g. SEM, SD) and the test used to calculate p-values is indicated in the respective figure legends (also for EV figures and all those in an Appendix). Please also check that all the p-values are explained in the legend, and that these fit to those shown in the figure. Please provide statistical testing where applicable. Please avoid the phrase 'independent experiment', but clearly state if these were biological or technical replicates. Please also indicate (e.g. with n.s.) if testing was performed, but the differences are not significant. In case n=2, please show the data as separate datapoints without error bars and statistics. See also: <http://www.embopress.org/page/journal/14693178/authorguide#statisticalanalysis>

9) Please also note our reference format: <http://www.embopress.org/page/journal/14693178/authorguide#referencesformat>

10) We updated our journal's competing interests policy in January 2022 and request authors to consider both actual and perceived competing interests. Please review the policy <https://www.embopress.org/competing-interests> and update your competing interests if necessary. Please name this section 'Disclosure and Competing Interests Statement' and put it after the Acknowledgements section.

11) We now use CRediT to specify the contributions of each author in the journal submission system. CRediT replaces the author contribution section. Please use the free text box to provide more detailed descriptions and do not provide your final manuscript text file with an author contributions section. See also our guide to authors:
<https://www.embopress.org/page/journal/14693178/authorguide#authorshipguidelines>

12) We would encourage you to use 'Structured Methods', our new Materials and Methods format. According to this format, the Materials and Methods section should include a Reagents and Tools Table (listing key reagents, experimental models, software, and relevant equipment and including their sources and relevant identifiers), uploaded as separate file, followed by a Methods and Protocols section in which we encourage the authors to describe their methods using a step-by-step protocol format with bullet points, to facilitate the adoption of the methodologies across labs. More information on how to adhere to this format as well as downloadable templates (.doc or .xls) for the Reagents and Tools Table can be found in our author guidelines (section 'Structured Methods'):

Please order the manuscript sections like this, using these names:

Title page - Abstract - Keywords - Introduction - Results - Discussion - Methods - Data availability section - Acknowledgements - Disclosure and Competing Interests Statement - References - Figure legends - Expanded View Figure legends

I look forward to seeing a revised version of your manuscript when it is ready. Please let me know if you have questions or comments regarding the revision.

Yours sincerely,

Referee #1:

Here, Van Gelder et al. demonstrate that in NK cells, IL-12 and IL-2 work synergistically to promote expression of *Irfng* mRNA and protein. They report a novel mechanism through which IL-2 (or PMA treatment) promote *Irfng* splicing in an NFκB-dependent fashion. Through finding that NK cells integrate two signals to coordinate maturation of *Irfng* pre-mRNAs, this study furthers our appreciation for how cells employ post-transcriptional regulatory mechanisms to mount balanced immune responses.

The manuscript is extremely well-written and the figures are beautifully formatted. Data supporting the model of IL-2/NFκB-dependent enhancement of *Irfng* splicing are strong. Experiments in Figs. 3 and 4 that help generalize their findings beyond the IL-12/IL-2 paradigm and specifically implicate NFκB activation in promoting *Irfng* splicing are notable contributions to the field.

However, the conclusion that altered *Irfng* splicing is responsible for changes to IFN-γ protein abundance is ill-supported by their data. The overall abundance of unspliced *Irfng* RNA relative to the total pool of mature *Irfng* mRNA, as measured by bioanalyzer or agarose gel electrophoresis, is very low. It is difficult to reconcile how such a small increase in splicing efficiency could be responsible for the 5-fold increase in protein reported in IL-12 + IL-2 treated cells-when cells in the two conditions appear to express very similar amounts of fully-processed *Irfng* mRNA. It is likely that other changes to IL-12 + IL-2 treated cells promote IFN-γ translation. I respect that they tried to control for overall enhanced translation in their puromycin incorporation assay. I do not think they need to figure out the precise mechanism through which IFN-γ protein abundance is enhanced in IL-12 + IL-2 treated cells. I do, however, think that language connecting the splicing phenotype to protein abundance needs to be toned down and potential alternative explanations should be laid out in the Discussion. This is particularly notable in the title "Interleukin-2 mediated NF-κB-dependent mRNA splicing induces acute interferon gamma protein production." If the authors insist on leaning into this splicing → protein abundance idea, they should include experimental evidence that modulating *Irfng* splicing can alter IFN-γ protein output in cells that are not treated with IL-12+IL-2.

Throughout the paper, the authors tend to use language that invokes kinetics, whereby the reliance on NFκB activation for *Irfng* intron removal slows acute accumulation/release of IFN-γ. Their data more accurately shows that at 6h, NK cells treated with IL-12 alone make very little IFN-γ protein, but NK cells treated with IL-12+IL-2 make lots of IFN-γ protein. If they want to comment on kinetics, they should follow IFN-γ accumulation in IL-12 v. IL-12 + IL-2 treated cells over a longer time course.

Minor errors

Some "gammas" are y's in the Figures.

In expanded Fig. 2B blue arrows are misaligned on first gel.

Referee #2:

Interferon-gamma (IFN γ) is an important cytokine whose expression control is vital for suppression of disease inducing inflammatory responses. To this end, the regulation of IFN γ expression has been proposed primarily through transcription control, with post-transcriptional regulation of IFN γ being less investigated. To this end, post transcriptional regulation of mRNAs and pre-mRNAs through splicing, RNA degradation, nuclear export, and secondary RNA structures are very relevant and underappreciated. For example, a large number of chromatin associated noncoding RNAs control gene expression of mRNAs expressed from nearby placed genes and their activity is regulated post transcriptionally by degradation via the RNA exosome complex. Similar local gene expression control via post transcriptional splicing is very relevant but remain under appreciated.

In this context, here Van Gelder et al report the expression of IFN γ is regulated via a post transcriptional splicing mechanism. Here, the authors report that when NK cells are treated with IL2 and IL12, IFN γ transcript is spliced to form mature mRNA with a concomitant increase in IFN γ protein. IL2 promotes IFN γ mRNA splicing, independent of transcription, but in a NF κ B dependent fashion. The conclusions are supported with very well designed experiments, the significance of results are nicely demonstrated and the experiments are properly controlled.

The manuscript is well written. The observation that IL2 signaling drives NF κ B mediated activation of IFN γ splicing is a novel and this the study should be well followed in the IFN field. I propose publication of this study. There are a couple of points the authors could consider during revisions.

1. In Fig. 1F, the kinetics of decrease of IFN γ mRNA following IL2 and IL12 treatment is quite comparable after 2 hrs onwards. Controls of some other mRNAs that are not IL2 and IL12 sensitive could be provided.
2. Is this a NK cell specific mechanism or does IFN γ mRNA expression via splicing occur in other immune cells. This could be considered for discussion.

Referee #3:

In this study, Van Gelder et al has proposed a new mechanism that promotes the expression of IFN γ mRNA via IL-2 mediated enhancement of IFN γ post-transcriptional splicing. While experiments were well designed and executed, there remain doubts concerning the validity and significance of this proposed mechanism.

Major comments:

1. Based on Figures 2B and S2B, the levels of nuclear-detained intron-retaining IFN γ transcripts were relatively low (presumably only 10% or less) compared to the spliced IFN γ transcripts. There wasn't an obvious switch of unspliced to spliced IFN γ isoform following IL2 treatment, making it difficult to believe that this post-transcriptional splicing is a major mechanism that induces acute IFN γ production. Despite not reaching statistical significance, the relative levels of nascent IFN γ did increase post IL-2 treatment (Figure S1C). Perhaps this will become significant if more replicates were included. Increased in the stability of IFN γ mRNA (Figure 1F) may be attributable to other factors (e.g. RNA modification). More evidence is needed to support the claim that this proposed post-transcriptional splicing mechanism is enhancing IFN γ expression independent of other mechanisms. At best, IL2 mediated enhancement of IFN γ intron excision may be involved in fine-tuning IFN γ mRNA levels.
2. Following on comment #1, the decrease in the relative levels of intron-retaining IFN γ after IL-2 treatment shown in Figure 2D needs to be considered in relation to the total amount of intron-retaining IFN γ prior to IL-2 treatment. This set of data was not provided. If there were only 10% or less intron-retaining IFN γ , how can the additional splicing of this small amount of intron-retaining IFN γ contribute to the marked increase in IFN γ mRNA post IL-2 treatment shown in Figure 1?
3. Similarly, for the mechanistic insights involving NF- κ B, it is important to consider the amount of intron-retaining IFN γ that are spliced to produce mature IFN γ . Even if it is true that the process occurs in an NF- κ B dependent manner, the significance of this mechanism is uncertain if it only regulates the splicing of a relatively small amount of IFN γ mRNA.

Minor:

1. Page 4: Fig S1G should be FigS1F.
2. Page 5: The claim that IL-2 mediated post-transcriptional splicing of IFN γ is not ubiquitous is an overstatement without a global profiling of splicing changes consequent to IL-2 treatment.
3. Page 5: The sentence "After stimulating IFN γ transcription with IL-12 (Fig 3B). Fig 3B doesn't seem to be the correct Figure to refer to.
4. Page 5: The claim that there is complete lack of nascent transcription (Fig 3F , Fig S4D) is not convincing because there is no measurement of nascent RNA levels.

Reponses to Reviewer Comments

We thank all reviewers for their thorough and thoughtful feedback and have incorporated their suggestions into our revised manuscript. Each comment is addressed point by point within this document.

The major revisions in this manuscript are expansion of the discussion and moderation of the language surrounding the contribution of intron retention to the synergistic IFN γ production that occurs in NK stimulated with IL-12 and IL-2. We modified our title, toned verbiage, and developed our discussion to include alternate mechanisms of post-transcriptional control that may account for the large observed increases of IFN γ protein. Line numbers of the revised manuscript corresponding to these changes are noted within our point-by-point responses.

In addition, we have added one primary figure and five supplemental figures to the manuscript, all of which align with findings from our initial submission.

Figure R1 (Revised Paper Figure 2E) indicates a strong kinetic correlation between initiation of *IFNG* mRNA splicing and the synergistic increase in IFN γ protein production during stimulation IL-12 and IL-2. This evidence strengthens the link between intronic splicing and protein induction by demonstrating that these events occur on a uniform timescale, beginning at 6 hours.

Figure R2 (Revised Paper Expanded View Figure 1D-F) demonstrates that the post-transcriptional stabilization effect of IL-2 is not universal, as stability of several control mRNAs is not affected in the presence versus absence of IL-2 stimulation.

Figure R3 (Revised Paper Expanded View Figure 2E) shows that nascently transcribed *IFNG* mRNA is spliced to a similar extent between 5- and 6-hours IL-12 or IL-12 + IL-2 stimulation as total *IFNG* mRNA over the course of 6 hours stimulation. This data suggests that intron containing *IFNG* mRNA may make up a relatively larger portion of the total pool of *IFNG* mRNA than is suggested by the data in Figures 2B and S2B.

Figure R4 (Revised Paper Expanded View Figure 2D) establishes the relative proportion of intron retention in IL-12 or IL-12 + IL-2 stimulated cells as compared with mock treatment, proving that a significant proportion of *IFNG* mRNA contains introns during IL-12 stimulation.

We believe that our revisions to this paper adequately address our reviewers' concerns and appropriately relate the impact of our findings to the field of post-transcriptional regulation of inflammatory mediators.

Referee #1:

Here, Van Gelder et al. demonstrate that in NK cells, IL-12 and IL-2 work synergistically to promote expression of Ifng mRNA and protein. They report a novel mechanism through which IL-2 (or PMA treatment) promote Ifng splicing in an NFkB-dependent fashion. Through finding that NK cells integrate two signals to coordinate maturation of Ifng pre-mRNAs, this study furthers our appreciation for how cells employ post-transcriptional regulatory mechanisms to mount balanced immune responses.

The manuscript is extremely well-written and the figures are beautifully formatted. Data supporting the model of IL-2/NFkB-dependent enhancement of Ifng splicing are strong. Experiments in Figs. 3 and 4 that help generalize their findings beyond the IL-12/IL-2 paradigm and specifically implicate NFkB activation in promoting Ifng splicing are notable contributions to the field.

We thank Reviewer 1 for their encouraging comments regarding our manuscript and for their thoughtful feedback.

However, the conclusion that altered Ifng splicing is responsible for changes to IFN- γ protein abundance is ill-supported by their data. The overall abundance of unspliced Ifng RNA relative to the total pool of mature Ifng mRNA, as measured by bioanalyzer or agarose gel electrophoresis, is very low. It is difficult to reconcile how such a small increase in splicing efficiency could be responsible for the 5-fold increase in protein reported in IL-12 + IL-2 treated cells-when cells in the two conditions appear to express very similar amounts of fully-processed Ifng mRNA. It is likely that other changes to IL-12 + IL-2 treated cells promote IFN- γ translation. I respect that they tried to control for overall enhanced translation in their puromycin incorporation assay. I do not think they need to figure out the precise mechanism through which IFN- γ protein abundance is enhanced in IL-12 + IL-2 treated cells. I do, however, think that language connecting the splicing phenotype to protein abundance needs to be toned down and potential alternative explanations should be laid out in the Discussion. This is particularly notable in the title "Interleukin-2 mediated NF-kB-dependent mRNA splicing induces acute interferon gamma protein production." If the authors insist on leaning into this splicing \rightarrow protein abundance idea, they should include experimental evidence that modulating Ifng splicing can alter IFN- γ protein output in cells that are not treated with IL-12+IL-2.

We agree that the language surrounding the contribution of splicing to the overall increase in IFN γ protein can be moderated, as we have not explicitly ruled out that other post-transcriptional mechanisms may contribute to modulation of protein production. With this in mind, we edited our verbiage throughout the manuscript (see Lines 36, 39, 153, 241 for key examples) and ensured we accounted for additional mechanisms beyond those explored by our study. Additionally, we altered the title of our manuscript; it now reads: "Interleukin-2 mediated NF-kB-dependent mRNA splicing *modulates* interferon gamma protein production." Our discussion has been expanded to elaborate on other potential mechanisms of post-transcriptional regulation that may alter protein production, including m6A RNA modifications that may result in increased mRNA

translation (Lines 246-257). We hope the reviewer will find these revisions appropriate solutions to their central concerns.

Throughout the paper, the authors tend to use language that invokes kinetics, whereby the reliance on NFkB activation for *Ifng* intron removal slows acute accumulation/release of IFN- γ . Their data more accurately shows that at 6h, NK cells treated with IL-12 alone make very little IFN- γ protein, but NK cells treated with IL-12+IL-2 make lots of IFN- γ protein. If they want to comment on kinetics, they should follow IFN- γ accumulation in IL-12 v. IL-12 + IL-2 treated cells over a longer time course.

We agree that the use of kinetics-oriented language regarding splicing is not effective without a relevant time-course demonstrating the relationship between splicing and protein production. We have therefore completed a 7.5 time-course with timepoints every 1.5 hours, quantifying both *IFNG* mRNA splicing and protein expression via ELISA at each point. Our data, which we have added as **Figure R1** (corresponding manuscript Figure 2E) strongly correlates increased splicing with synergistic protein production during IL-12 + IL-2 stimulation across time (Lines 128-135). We hope that this experiment and our changes in verbiage will satisfy Reviewer 1's concerns.

Figure R1 (Fig 2E): SYBR qPCR analysis of *IFNG* intron retention in NK92 whole cell lysates during a time course of IL-12 (10ng/mL) or IL-12 + IL-2 (100U/mL) stimulation for 7.5 hours, normalized to the IL-12 stimulation condition and to the *IFNG* 5'UTR within each timepoint. Plotted on the same graph as IFN γ protein production as measured from ELISA of supernatants also collected over the 7.5 hours treatment course. Data is mean \pm SEM of 3 biological replicates analyzed by two-way ANOVA with Turkey's test for multiple comparisons * $p \leq 0.05$, ** $p \leq 0.01$.

Minor errors

Some "gammas" are y's in the Figures.

We have corrected and substituted gamma for y's where appropriate.

In expanded Fig. 2B blue arrows are misaligned on first gel.

This error has been corrected

Referee #2:

Interferon-gamma (IFN γ) is an important cytokine whose expression control is vital for suppression of disease inducing inflammatory responses. To this end, the regulation of IFN γ expression has been proposed primarily through transcription control, with post-transcriptional regulation of IFN γ being less investigated. To this end, post transcriptional regulation of mRNAs and pre-mRNAs through splicing, RNA degradation, nuclear export, and secondary RNA structures are very relevant and underappreciated. For example, a large number of chromatin associated noncoding RNAs control gene expression of mRNAs expressed from nearby placed genes and their activity is regulated post transcriptionally by degradation via the RNA exosome complex. Similar local gene expression control via post transcriptional splicing is very relevant but remain under appreciated.

In this context, here Van Gelder et al report the expression of IFN γ is regulated via a post transcriptional splicing mechanism. Here, the authors report that when NK cells are treated with IL2 and IL12, IFN γ transcript is spliced to form mature mRNA with a concomitant increase in IFN γ protein. IL2 promotes IFN γ mRNA splicing, independent of transcription, but in a NF κ B dependent fashion. The conclusions are supported with very well designed experiments, the significance of results are nicely demonstrated and the experiments are properly controlled.

The manuscript is well written. The observation that IL2 signaling drives NF κ B mediated activation of IFN γ splicing is a novel and this the study should be well followed in the IFN field. I propose publication of this study. There are a couple of points the authors could consider during revisions.

We thank Reviewer 2 for their support of our manuscript and have addressed their minor concerns with several experiments, as well as an expansion of the discussion section as detailed below.

1. In Fig. 1F, the kinetics of decrease of IFN γ mRNA following IL2 and IL12 treatment is quite comparable after 2 hrs onwards. Controls of some other mRNAs that are not IL2 and IL12 sensitive could be provided.

We investigated the transcription factor *RELC*, knockdown of which we have shown does not affect the expression of *IFNG*, in the context of IL-12 + IL-2 treatment, and quantified degradation of its transcript. We show in **Fig R2** (now Expanded View Figure 1D), that stimulation with IL-2 after treatment with Actinomycin D, which halts transcriptional processes, does not affect degradation of *RELC* mRNA (Lines 105-109). This finding suggests that IL-2 mediated mRNA stabilizing effects are not universal. We also investigated degradation of the RNA binding protein *HNRNPU* and the housekeeping gene *GAPDH* and found that IL-2 did not affect longevity of either transcript (Revised manuscript Fig EV1E, EV1F).

Figure R2 (Fig EV1D-F): NK92 cells were stimulated with IL-12 (10ng/mL) for 3 hours, then all nascent transcription halted with ActD (5ug/mL) and IL-2 (100u/mL) stimulation added or omitted for 3 hours. Time course of *RELC*, *HNRNPU*, and *GAPDH* mRNA stability with or without IL-2 stimulation, normalized to *RELC*, *HNRNPU*, and *GAPDH* mRNA levels at 3h IL-12 treatment before addition of ActD, respectively.

2. Is this a NK cell specific mechanism or does IFN γ mRNA expression via splicing occur in other immune cells. This could be considered for discussion.

We agree with the reviewer that the applicability of our mechanism to other *IFNG* producing cells is certainly of interest. We expanded our discussion accordingly to include discourse on CD4+ T cells, the other major *IFNG* expressing cell type during the antimicrobial response, detailing how we imagine a similar post-transcriptional mechanism might arise in the context of T-cell receptor and autocrine IL-2 signaling. These edits can be found on Lines 285-293 of the revised manuscript.

Referee #3:

In this study, Van Gelder et al has proposed a new mechanism that promotes the expression of IFN γ mRNA via IL-2 mediated enhancement of IFN γ post-transcriptional splicing. While experiments were well designed and executed, there remain doubts concerning the validity and significance of this proposed mechanism.

Major comments:

1. Based on Figures 2B and S2B, the levels of nuclear-detained intron-retaining IFN γ transcripts were relatively low (presumably only 10% or less) compared to the spliced IFN γ transcripts. There wasn't an obvious switch of unspliced to spliced IFN γ isoform following IL2 treatment, making it difficult to believe that this post-transcriptional splicing is a major mechanism that induces acute IFN γ production. Despite not reaching statistical significance, the relative levels of nascent IFN γ did increase post IL-2 treatment (Figure S1C). Perhaps this will become significant if more replicates were included. Increased in the stability of IFN γ mRNA (Figure 1F) may be attributable to other factors (e.g. RNA modification). More evidence is needed to support the claim that

this proposed post-transcriptional splicing mechanism is enhancing IFNG expression independent of other mechanisms. At best, IL2 mediated enhancement of IFNG intron excision may be involved in fine-tuning IFNG mRNA levels.

We thank Reviewer 3 for their thoughtful insights regarding the impact of our findings. We understand the reviewer's central concern: that the amount of observed intron splicing does not account for the large changes in protein and total RNA induced upon treatment with IL-2. To first address this concern, we have expanded our discussion (Lines 246-257) to detail potential other mechanisms by which IL-2 might mediate large increases in transcript and protein—Reviewer 1 also expressed a desire for increased discussion on this topic. We agree that while splicing plays a role in the regulation of *IFNG* transcript and expression based on our study, additional mechanisms, such as enhanced post-transcriptional loading onto polysomes for increased translation, could play significant roles in our phenotype. We hope our expanded discussion will satisfy the remaining questions of the Reviewer in this regard.

We additionally agree that the findings in Figure 1E/S1C, while not statistically significant, do trend toward IL-2 increasing the nascent transcription of *IFNG* mRNA, and therefore suggest that IL-2 may contribute to downstream protein production partially through prompting increased transcription. We have expanded the results text to reflect this possibility (Lines 75-80, 97-99).

To provide more evidence that splicing is linked with protein production, we kinetically correlated *IFNG* mRNA processing with heightened IFN γ protein expression during the combination of IL-12 and IL-2 stimulation (Lines 128-135). Over a time-course of 7.5 hours, we found that significant *IFNG* mRNA splicing, beginning at 6 hours post stimulation, coincided exactly with the induction of synergistic IFN γ protein production, or when IFN γ expression in IL-12 versus IL-12 + IL-2 stimulated cells widened appreciably (**Fig R1**, corresponding manuscript Fig 2E). We hope the reviewer agrees that this data strengthens the association between mRNA processing and robust IFN γ expression and satisfies apprehensions when taken in combination with our augmented discussion of other post-transcriptional mechanisms modulating IFN γ production.

Regarding intron retaining transcripts comprising relatively little of the total *IFNG* mRNA pool in Figure 2B/EV2B, we propose that saturation of the spliced pool of *IFNG* mRNA occurs over the course of 6 hours stimulation dilutes the population of intron containing *IFNG* mRNA. We postulate that this effect is magnified due to rapid turnover of unprocessed, intron-containing mRNA, further increasing the relative size of the total *IFNG* mRNA pool compared with the portion that contains introns. Thus, though nascently transcribed RNA may have significant intron retention, it would appear on a gel as a smaller percentage of the *IFNG* mRNA population than it might comprise at a given moment. To prove this concept, we conducted an experiment to probe the relative abundance of intron containing versus spliced *IFNG* mRNA in nascently transcribed *IFNG* mRNA, under the IL-12 versus IL-12 + IL-2 stimulation conditions using 4SU labelling (**Figure R3**). We found after 6 hours IL-12 + IL-2 stimulation, nascently transcribed RNA is spliced to a similar extent as total RNA. This finding suggests that

intron retaining transcripts may represent a relatively larger portion of the total *IFNG* mRNA pool than is observed over 6 hours, as the amount of spliced RNA increases with *IFNG* mRNA induction throughout the stimulation period. (Line 140-145).

Figure R3 (Fig EV2E): SYBR qPCR analysis of *IFNG* intron retention in nascently transcribed or total mRNA from whole cell lysates stimulated with IL-12 or IL-12 + IL-2. Cells were pulsed with 4SU between 5- and 6-hours stimulation. Intron expression normalized to *IFNG* 5'UTR, representing total mature plus unspliced *IFNG* mRNA as quantified by amplification of the region spanning the 5'UTR into the coding region of Exon 1 as control. Data shown is mean \pm SEM of 3 biological replicates, paired T test with Holm-Šidák method for multiple corrections. * $p \leq 0.05$, n.s. is not significant

2. Following on comment #1, the decrease in the relative levels of intron-retaining IFNG after IL-2 treatment shown in Figure 2D needs to be considered in relation to the total amount of intron-retaining IFNG prior to IL-2 treatment. This set of data was not provided. If there were only 10% or less intron-retaining IFNG, how can the additional splicing of this small amount of intron-retaining IFNG contribute to the marked increase in IFNG mRNA post IL-2 treatment shown in Figure 1?

We thank Reviewer 3 for their comment and recognize the concern of the relative contribution of the splicing phenotype to the induction of synergistic IFN γ protein. To this end, we have reanalyzed an existing data set by normalizing *IFNG* mRNA splicing that occurs during IL-12 and IL-12 + IL-2 splicing to intron retention quantified in the mock treated condition (**Figure R4**, manuscript Fig EV2D, Lines 126-128). As we hope the reviewer can appreciate, in the IL-12 stimulated condition, 35-50% of intron containing transcripts remain, while after just 3 hours additional IL-2 treatment, *IFNG* intron retention drops to 10% or lower in Intron 1, showing that a significant portion of IL-12 stimulated cells contain transcripts with their introns intact. Throughout our manuscript, we do not normalize to the mock treated condition primarily because there is little *IFNG* mRNA produced in unstimulated NK cells, resulting in high CT values and large variability of transcript amplification.

We do agree, however, that our initial submission may lend more credence to splicing than is appropriate given our findings and have thus edited our verbiage to tone down

the significance of the splicing contribution (see Lines 36, 39, 153 for examples of text edits that span the manuscript). We have gone so far as to edit the title, removing that splicing induces “acute” protein production in favor of the more moderate “modulates” protein production. Additionally, we have expanded upon our discussion to include further methods that may account for the discrepancy given that splicing likely does not singlehandedly result in synergistic protein induction (Line 246-257).

Figure R4 (Fig EV2D): SYBR qPCR analysis of *IFNG* intron retention in whole cell lysates during stimulation with IL-12 for 3h with or without 3h IL-2 subsequent stimulation. Each stimulation condition normalized to mock treatment condition. Intron expression normalized to *IFNG* 5’UTR, representing total mature plus unspliced *IFNG* mRNA as quantified by amplification of the region spanning the 5’UTR into the coding region of Exon 1 as control. Data shown is mean ± SEM of 3 biological replicates, paired T test with Holm-Šídák method for multiple corrections. * $p \leq 0.05$, ** $p \leq 0.01$

3. Similarly, for the mechanistic insights involving NF-KB, it is important to consider the amount of intron-retaining IFNG that are spliced to produce mature IFNG. Even if it is true that the process occurs in an NF-KB dependent manner, the significance of this mechanism is uncertain if it only regulates the splicing of a relatively small amount of IFNG mRNA.

We acknowledge that if intron retention is not the only post-transcriptional mechanism modulating synergistic induction of IFN γ protein, the significance of NF-kB is less broadly applicable in modulating control of protein expression. However, we still believe our finding is worthy to report, as NF-kB signaling is not classically associated with post-transcriptional control and is also a non-canonical pathway downstream of the IL-2 receptor. We therefore feel validated in including this data as a novel means of post-transcriptional regulation of IFN γ protein.

Minor:

1. Page 4: Fig S1G should be FigS1F. We have corrected this error.
2. Page 5: The claim that IL-2 mediated post-transcriptional splicing of IFNG is not ubiquitous is an overstatement without a global profiling of splicing changes consequent to IL-2 treatment.

We have edited our verbiage to tone down this statement and hope the reviewer will find this appropriate (Line 104-109).

3. Page 5: The sentence "After stimulating IFNG transcription with IL-12 (Fig 3B). Fig 3B doesn't seem to be the correct Figure to refer to. We have addressed this misdirection to the figure.

4. Page 5: The claim that there is complete lack of nascent transcription (Fig 3F, Fig S4D) is not convincing because there is no measurement of nascent RNA levels. We have removed this statement from the manuscript since we have not provided explicit evidence that treatment with ActD completely halts nascent transcription. (Line 192)

Dear Dr. Savan,

Thank you for the submission of your revised manuscript to our editorial offices. I have now received the report from the two referees that were asked to re-evaluate the study, you will find below. As you will see, both referees now fully support the publication of the study in EMBO reports. Referee #2 already supported publication of the previous version.

Before I can proceed with formal acceptance, I have these editorial requests I ask you to address in a final revised manuscript:

- Please order the manuscript sections like this, using these names:

Title page - Abstract - Keywords - Introduction - Results & Discussion - Methods - Data availability section (DAS) - Acknowledgements (including funding information) - Disclosure and Competing Interests Statement - References - Figure legends - Expanded View Figure legends

- Please remove the word 'sequencing' from the DAS. I would suggest: 'No primary large datasets have been generated or deposited for this study'.

- The nomenclature (legend and file names) for the EV figures is wrong. It should be 'Figure EVx', not 'Expanded View Figure x'.

- Please provide titles for the EV figures in their legends.

- The resolution for the Western blots is very low and they are very pixelated. Please provide the Western blots with a higher resolution.

- Please make sure that the number "n" for how many independent experiments were performed, their nature (biological versus technical replicates), the bars and error bars (e.g. SEM, SD) and the test used to calculate p-values is indicated in the respective figure legends. Please also check that all the p-values are explained in the legend, and that these fit to those shown in the figure. Please provide statistical testing where applicable. Please avoid the phrase 'independent experiment', but clearly state if these were biological or technical replicates. Please also indicate (e.g. with n.s.) if testing was performed, but the differences are not significant. In case n=2, please show the data as separate datapoints without error bars and statistics. See also: <http://www.embopress.org/page/journal/14693178/authorguide#statisticalanalysis>

If n<5, please show single datapoints for diagrams. Moreover:

- Please provide the exact p values in the legends of figures 1A-D, F; 2D-F; 3A-F, H-I; 4A-G; EV1 B, EV2 C, D, E; EV3 E; EV4 C-F.

- Please add to each legend (main, EV and Appendix figures, where applicable) a 'Data Information' section explaining the statistics used or providing information regarding replicates and scales. See:

- Please make sure that all figure panels are called out separately and sequentially. Presently, there seems to be no separate callouts for Fig 3I. There is also a callout to Figure S2B. Please check. Moreover, there are callouts to Expanded View Tables 1 and 2, and to Table S1. Please update these (see also below).

- All Materials and Methods need to be described in the main text using our 'Structured Methods' format, which is required for all research articles. According to this format, the Methods section should include a Reagents and Tools Table (listing key reagents, experimental models, software, and relevant equipment and including their sources and relevant identifiers), uploaded as separate file, followed by a Methods section in which we encourage the authors to describe their methods using a step-by-step protocol format with bullet points, to facilitate the adoption of the methodologies across labs. More information on how to adhere to this format as well as downloadable templates (.doc) for the Reagents and Tools Table can be found in our author guidelines (section 'Structured Methods'):

<https://www.embopress.org/page/journal/14693178/authorguide#structuredmethods>

Please add the primer information (presently uploaded as Table EV1) to this table (and remove Table EV1). Please add or update callouts to the reagents and tools table.

- Please make sure that all the funding information is also entered into the online submission system and that it is complete and similar to the one in the acknowledgement section of the manuscript text file. Presently, the grants ITHS (5TL1TR002318-04), UW Immunology Richard Titus fellowship, and grants from the intramural research programs of the National Cancer Institute, Center for Cancer Research, Cancer Innovation Laboratory are missing in the submission system. Please check.

- Thank you for providing the requested source data (SD). Please upload this separated, with one folder per figure (with all files for one figure in one folder and ZIPed).

In addition, I would need from you uploaded separately:

Best,

Referee #1:

The authors have done a commendable job responding to reviews. I look forward to seeing this important work published in EMBO Reports.

Referee #3:

This reviewer would like to commend the authors for addressing the previous queries thoroughly. The addition of new data to show that IFGN protein levels increased significantly with reduced intron retention over time has strengthened the results. The authors have also toned down the claims and provide further insights into other potential mechanisms that may act synergistically with intron retention to control IFGN production consequent to IL-2 stimulation.

All editorial and formatting issues were resolved by the authors.

Ram Savan
UW
Center for Innate Immunity and Immune Diseases
750 Republican Street, E307
Seattle, WA 98109
United States

Dear Dr. Savan,

I am very pleased to accept your manuscript for publication in the next available issue of EMBO reports. Thank you for your contribution to our journal.

Yours sincerely,
